

# Impact of Arctic Amplification variability on the chemical composition of the snowpack in Svalbard

Azzurra Spagnesi[a,b], Elena Barbaro[a,b] *, Matteo Feltracco[a,b], Federico Scoto[c,b], Marco Vecchiato[b], Massimiliano Vardè[a], Mauro Mazzola[a], François Burgay[d,e], Federica Bruschi[f], Clara Jule Marie Hoppe[g], Allison Bailey[g], Andrea Gambaro[a,b], Carlo Barbante[a,b], Andrea Spolaor[a,b]

[a] Institute of Polar Sciences - National Research Council of Italy (ISP-CNR), Via Torino 155, 30172, Venice, Italy

[b] Department of Environmental Sciences, Informatics and Statistics, Ca' Foscari University of Venice, Via Torino 155, 30172, Venice, Italy

[c] Institute of Atmospheric Sciences and Climate - National Research Council of Italy (ISAC-CNR), Campus Ecotekne, Lecce, 73100, Italy

[d] Laboratory of Environmental Chemistry (LUC), Paul Scherrer Institut (PSI), Villigen, 5232, Switzerland

[e] Oeschger Centre for Climate Change Research, University of Bern, Bern, 3012, Switzerland

[f] Department of Chemistry, Biology and Biotechnology, University of Perugia, Via dell'Elce di Sotto 8, 06123, Perugia, Italy

[g] Alfred Wegener Institute, Helmholtz Centre for Polar and Marine Research, 27570 Bremerhaven, Germany

Corresponding: Elena Barbaro (elena.barbaro@cnr.it)

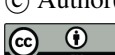



## Abstract

Arctic Amplification (AA) is leading to significant glacier ice melting, rapid sea ice decline, and alterations in atmospheric and geochemical processes in the Arctic regions, with consequences on the formation, transport, and chemical composition of aerosols and seasonal snowpack. Svalbard is particularly exposed to the AA, thus represents a relevant site in the Arctic to evaluate changes in local environmental processes contributing to the seasonal snow chemical composition. Sampling campaigns were conducted from 2018 to 2021 at the Gruvebadet Snow Research Site in Ny-Ålesund, in the North-West of the Svalbard Archipelago. During the investigated years, interannual variability of ionic and elemental impurities in surface snowpack has been associated to an alternation between relative warm years (2018-19, 2020-21), typical of the Arctic Amplification (AA) period, and relatively cold years (2019-20), more similar to the pre-AA conditions. Our results indicate that the concentration of impurities during the colder sampling season is strongly dependent on the production of sea spray related aerosol, likely deriving by a larger extension of sea ice, and drier, windy conditions. Our findings were therefore linked to the presence of sea ice in the Kongsfjorden in March 2020, and more generally around Spitsbergen, resulting from the exceptional occurrence of a strong and cold wintry stratospheric polar vortex and unusual AO index positive phase. By comparing the snow chemical composition of the 2019-20 season with 2018-19 and 2020-21, we present an overview of the possible impact of AA on the Svalbard snowpack, and the related change in the aerosol production process.

## 1. Introduction

Chemical analysis of surface Arctic snow and ice can provide valuable comprehension of the composition of Arctic aerosols, its deposition, and exchange processes (Lai et al., 2017), which may be variously influenced by the Arctic Amplification (AA), a non-linear increase in near-surface air temperatures observed from 1975 to date (Chylek et al., 2022). AA is recognized as an inherent characteristic of the changing global climate system, with multiple intertwined causes operating on a spectrum of spatial and temporal scales. These include, but are not limited to, changes in sea ice extent that impact heat fluxes between the ocean and the atmosphere, and water vapor that alters longwave radiation (Serreze and Barry, 2011). The Svalbard Archipelago is particularly affected by AA due to the relatively low altitude of its main ice fields and its geographical location in the higher North Atlantic, which make the effect of AA more significant (Spolaor et al., 2024). Therefore, in the 21[st] century, predicting and characterizing climate change in Svalbard is particularly crucial, as changes in near-surface air temperature, precipitation, and sea ice extent occur at an extremely high

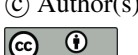



pace (Gjermundsen et al., 2020; Rantanen et al., 2022). The Svalbard region, located at the southern
edge of the seasonal Arctic sea ice zone, is characterized by a maritime climate with strong
temperature variations during winter (Hansen et al., 2014; Barbaro et al., 2021). In the Arctic winter,
the stratospheric polar jet fosters a high-atmospheric vorticity zone. This winter vortex typically acts
as a strong barrier for long-range transport of pollutants from mid-latitudes (Lawrence et al., 2020).
However, it occasionally allows warm southern air to penetrate the region (Schoeberl and Newman,
2015). Additionally, Svalbard frequently experiences intense cyclonic storms in autumn and winter,
which bring both heat and moisture from lower latitudes (Rinke et al., 2017). These intense
meteorological variations, generally linked with a weaker polar vortex (Sobota et al., 2020; Salzano
et al., 2023), favor long-range transport of aerosols to the archipelago, including pollutants from
continental sources (Stohl et al., 2006b; Yttri et al., 2014a; Vecchiato et al., 2024; D'Amico et al.,

74 2024).

Arctic snow captures dry and wet deposition and forms an archive that includes a range of seasonal
chemical species such as major ions and trace elements, as well as human-made pollutants emitted
into the Arctic atmosphere (Koziol et al., 2021). Ny-Ålesund is a well-monitored area and a natural
laboratory for complex system observations, ideal for exploring both long-range contaminants from
mid- to high-latitude regions of Eurasia and Canada (Nawrot et al., 2016; Song et al., 2022; Vecchiato
et al., 2024; D'Amico et al., 2024), and local inputs from both natural processes and human settlement
(Vecchiato et al., 2018). While previous research investigated the temporal and compositional aspects
of the Ny-Ålesund lower atmosphere (Stohl et al., 2006a; Eleftheriadis et al., 2009; Geng et al., 2010;
Zhan et al., 2014; Feltracco et al., 2020, 2021; Turetta et al., 2021), the chemistry of Arctic snow and
the exchange of inorganic species between cryosphere and atmosphere have been the subject of a
relatively small number of studies or of specific events (Dommergue et al., 2010; Spolaor et al., 2013,
2019; Barbante et al., 2017).
In this study, we evaluate the surface snow concentration of ionic ($Cl^-$, $Br^-$, $NO_3^-$, $SO_4^{2-}$, MSA, $Na^+$,
$NH4^+$, $K^+$, $Ca^{2+}$) and elemental impurities (Li, Be, Mg, Al, Ca, V, Cr, Mn, Fe, Co, Ni, Cu, Zn, As,
Se, Rb, Sr, Ag, Cd, Sb, Cs, Ba, Tl, Pb, Bi, U) for the snow seasons between 2018-2021, at the
Gruvebadet Snow Research Site (GSRS) location, 1 km far from Ny-Ålesund, where clean and
undisturbed snow conditions are guaranteed throughout the whole sampling season.
The differences in average meteorological and climatological conditions across the studied seasons
are analysed to assess how sea ice extent, polar vortex, and Arctic Oscillation (AO) conditions
influence the composition of surface snow in connection with the aerosol-producing and deposition
processes in Kongsfjorden.



## 2. Methodology

### 2.1 Sampling and processing

Three sampling campaigns were conducted in Svalbard between 2018 and 2021, covering the period from October to May according to the onset of the snowpack formation and melting.

During the first sampling campaign, carried out from October 4th, 2018 to May 10th, 2019, 114 surface snow samples were collected in a delimited snow field located ~ 100 m south of the "Dirigibile Italia Station" in Ny-Ålesund (78.92° N 11.93° E, Ny-Ålesund, Svalbard). The surface snow was sampled within the upper 3 cm, as this is the snow layer most influenced by the aerosol-cryosphere exchanges, and, in case of snowfall, by deposition (Spolaor et al., 2018, 2021b). This choice also minimised the effect of different physical snow conditions (density, crystal shape and size).

Concurrently, additional 133 snow samples were collected at the Gruvebadet Snow Research Site (GSRS) to evaluate the spatial variability with respect to the snow samples collected in Ny-Ålesund. The GSRS is a clean-area located about 1 km south of Ny-Ålesund, nearby the Gruvebadet Atmospheric Laboratory (GAL), dedicated to the chemical and physical monitoring of the seasonal snowpack (Scoto et al., 2023; Fig. S1). Throughout the season, the sampling resolution varied based on light conditions. During the polar night (from October to early March), snow sampling was carried out daily at Ny-Ålesund, and every 3-5 days at the GSRS. With the beginning of the polar day, daily sampling was conducted both in Ny-Ålesund and at the GSRS in March, and then continued only at the GSRS until the end of the snow season in June due to the lower contamination of the site, more distant from the fervent local activities. This sampling resolution overlap during March ensured a good comparison of results in both snow fields (Fig. S2).

Starting from the second campaign, snow sampling activities were conducted only at the GSRS site, since clean conditions of the field in Ny-Ålesund could not be guaranteed due to construction works. The snow sampling was carried out from October 26th, 2019 to May 25th, 2020, with a total of 107 samples collected. The surface snow layer was sampled every 3-5 days during the polar night (until February 24th, 2020), and daily from the beginning of the polar day until the end of the snow season.

Finally, during the third snow sampling campaign, lasting from October 27th, 2020 to June 15th, 2021, a weekly sampling was conducted at GSRS, with a total of 32 samples collected.

During snow sampling, the temperature and density of surface snow were measured, and the density of snow was calculated based on weighting a 100 cc cylinder. After collection, snow samples were melted, and two different aliquots were obtained and stored in separate vials. In a 1.5 mL polypropylene (PP) vial, 1 mL of sample was stored for ionic species, while another aliquot was stored in a 5 mL LDPE vials for trace elements analysis. PP vials designated to ionic species analysis





were previously sonicated for 30 min in UltraPure Water (UPW) (18 MΩ cm$^{-1}$ at 25 °C) for
decontamination. LDPE vial used for trace elements analysis were instead conditioned with $HNO_3$
2% and sonicated for 30 min. All sample aliquots were stored at -20°C in dark conditions and
transported to the Venice ISP-CNR laboratories.
Furthermore, seawater temperatures and salinity at 10 m depth were also monitored in Kongsfjorden
(Kb3; 78°57.228'N, 11°57.192'E) during 2019-2021 spring seasons, with data collected every 3-6
days (Assmy et al. 2023). Data was derived from Conductivity Temperature Depth (CTD) casts with
either a MiniSTD model SD-204 (SAIV A/S, Bergen, Norway) or a XR-620 CTD (RBR Ltd, Ottawa,
Canada). Combined casts of both instruments conducted in May 2020 and 2021 did not reveal
differences in temperature or salinity in the reported accuracy (two post comma digits).
*2.2 Analysis of ionic species*
The analysis of anionic species (Cl$^-$, Br$^-$, $NO_3^-$, $SO_4^{2-}$, MSA) was carried out using an ion
chromatograph (IC, Thermo Scientific Dionex™ ICS-5000, Waltham, MA, USA) coupled with a
single quadrupole mass spectrometer (MS, MSQ Plus™, Thermo Scientific, Bremen, Germany). The
separation was performed using an anionic exchange column (Dionex Ion Pac AS 19 2 mm ID × 250
mm length) equipped with a guard column (Dionex Ion Pac AG19 2 mm ID × 50 mm length). Sodium
hydroxide (NaOH), used as mobile phase, was produced by an eluent generator (Dionex ICS 5000EG,
Thermo Scientific). The NaOH gradient with a 0.25 mL min$^{-1}$ flow rate was: 0-6 min at 15 mM; 6-
15 min gradient from 15 to 45 mM; 15-23 min column cleaning with 45 mM; 23–33 min equilibration
at 15 mM. The injection volume was 100 μL. A suppressor (ASRS 500, 2 mm, Thermo Scientific)
removed NaOH before entering the MS source. The IC-MS operated with a negative electrospray
source (ESI) with a temperature of 500°C and a needle voltage of 3 kV. The other MS parameters are
reported by Barbaro et al. (2017). The same IC system was simultaneously used to determine cationic
species (Na$^+$, K$^+$, Ca$^{2+}$ and $NH_4^+$). However, Ca$^{2+}$ was not measured within the samples collected
during the second campaign due to instrumental limitations.
The separation occurred with a capillary cation-exchange column (Dionex Ion Pac CS19–4 mm 0.4
mm ID × 250 mm length), equipped with a guard column (Dionex Ion Pac CG19–4, 0.4 mm ID × 50
mm length), and the species were determined using a conductivity detector. Analytical blanks of
ultrapure water (> 18 MΩ cm) were included in the analysis, and the Method Detection Limit (MDL)
was set to 3 times the standard deviation of the blank values. Checks for accuracy were made using
certified multi-element standard solutions for anions (Cl$^-$, Br$^-$, $NO_3^-$, $SO_4^{2-}$, no. 89886-50ML-F, Sigma
Aldrich) and cations (Na$^+$, K$^+$, Ca$^{2+}$, no. 89316-50ML-F, Sigma Aldrich) at a concentration of 10 mg





L$^{-1}$ ± 0.2%. The analytical precision was quantified as the relative standard deviation (RSD) for
replicates (n > 3) of standard solutions and was always < 10% for each ion.
*2.3 Trace Elements analysis*
Twenty-six elements (Li, Be, Mg, Al, Ca, V, Cr, Mn, Fe, Co, Ni, Cu, Zn, As, Se, Rb, Sr, Ag, Cd, Sb,
Cs, Ba, Tl, Pb, Bi and U) were analyzed on samples previously melted and acidified to 2% v/v with
HNO$_3$ (UpA grade, Romil, UK) for 24 hours before analysis (Spolaor et al., 2018; Spolaor et al.,
2021a).
The analysis was performed using Inductively Coupled Plasma Mass Spectrometry (ICP-MS, iCAP
RQ, Thermo Scientific, US). The ICP-MS was equipped with an ASX-560 autosampler (Teledyne
Cetac Technologies), PolyPro PFE nebulizer, PFE cyclonic spray chamber thermostated at 2.7°C,
sapphire injector, quartz torch and Ni cones. The acquisition was performed at 1550 W of plasma RF
power in Kinetic Energy Discrimination (KED) – high matrix mode, using He as collision gas (4.3
mL min$^{-1}$). Instrument parameters were optimized for best sensitivity in the whole mass range,
minimum oxides (< 1%) and double charges (< 3%). Quantification was obtained by external
calibration with multi-elemental standards prepared in ultrapure water (18 MΩ cm$^{-1}$ at 25° C) with
2% v/v ultrapure grade HNO$_3$ (UpA grade, Romil, UK), with a combination of certified level multi-
elemental solutions IMS-102 and IMS-104 from UltraScientific. Analytical quality control was
performed by memory test blank (repeated analysis of ultrapure grade HNO$_3$ 2% v/v blank solution)
after each sample and calibration verification (repeated analysis of reference standards) every 11
samples. More details are found in Spolaor et al., 2021a.

*2.4 Transport modelling, sea ice, Kongsfjorden condition, and polar vortex*

The Lagrangian particle dispersion model HYSPLIT (Draxler, 1998; Stein et al., 2015) was used to
determine the source region of air masses over Ny-Ålesund. This model has previously been shown
to be an effective tool for the prediction of transport pathways into and within the Arctic and Antarctic
regions (Barbaro et al., 2015; Feltracco et al., 2021). The simulations were driven using
meteorological data from the Global Data Assimilation System (GDAS) one-degree archive, set the
top of the model at 10000 m and the height source equal to the GSRS altitude. Back-trajectories were
calculated every 6 h, with a propagation time of 120 h for each sampling period, as suggested in
previous studies on atmospheric circulation in the same site (Feltracco et al., 2021). This approach
was used to ensure an envelope working for all investigated tracers. The resulting multiple trajectories
were based on the screen-plot analyses of total spatial variance.



The Ice Service provided by the Norwegian Meteorological Institute (NIS) was employed to analyse the weather conditions via remotely sensed data and to generate ice charts of Svalbard, ice-edge information, and sea surface temperatures trends. Sea ice extent variability in Kongsfjorden was evaluated based on dataset made available by Gerland et al. (2022).

Differences between the sampling campaigns were evaluated through the NCEP/NCAR Reanalysis data from NOAA Physical Sciences Lab's daily composites tool, used to calculate the near-surface air temperatures across the Northern Hemisphere from October to May.

*2.5 Statistical procedures*

Results below the limit of detection were assumed to be equal to ½ of Method Determination Limit (MDL) prior to perform statistical analysis, to approximate their likely level based on the data distribution curve (best approximated as log-normal for most of the studied variables) (George et al., 2021).

The Wilcoxon test was applied on data from the 2018-19 sampling campaign conducted at Ny-Ålesund and Gruvebadet to determine whether the difference between the population median and the hypothesized median of surface snow contamination level was statistically significant. This model assumes that the data is sampled from two matched or dependent populations, and data is assumed to be continuous. Because it is a nonparametric test, it does not require a particular probability distribution of the dependent variable in the analysis. Furthermore, a Hierarchical Cluster Analysis (HCA) was performed using Ward's algorithm and Euclidean distances as clustering criteria, to determine the presence of some clusters and simplify the interpretation of the dataset.

## 3 Results

*3.1 Comparison between concentration trends at Gruvebadet and Ny-Ålesund*

The concentration variations between an undisturbed area in Ny-Ålesund village and GSRS sites were compared during the 2018-19 sampling campaign to better understand the effect of spatial variability between the two sampling sites. The concentration trends of $Na^+$, as sea salt tracer, Pb as anthropogenic species, and $Ca^{2+}$ as crustal tracer, are reported in Fig. S2, for both sampling sites. Although the difference in time resolution between sites is apparent in Fig. S2, the difference in concentration trends appears very low or negligible, with few isolated peaks for sea salt and crustal tracers present in the Ny-Ålesund record from November to February, following positive temperature anomalies and precipitation events (Fig. S2). Concordant Pb trends emerge at Ny-Ålesund and Gruvebadet, with highest concentrations observed from February to May.



To evaluate the differences in concentration range and spatial distribution of surface snow impurity
content, we applied the Wilcoxon test for the 2018-19 sampling period by comparing the distributions
for positive and negative differences of the ranks of their absolute values. At a significance level of
0.01, the two distributions from GSRS and Ny-Ålesund sites were not statistically different for all the
trace elements and most of the inspected ions.
For this reason, only the GSRS temporal trend has been considered throughout the manuscript,
referring to ionic loads (mg m$^{-2}$) instead of concentrations (ng g$^{-1}$), to highlight the seasonal trends of
specific tracers. The ionic load is calculated as ionic concentrations multiplied by the density and the
depth of sampled strata.
*3.2 Interannual trends of chemical species on the surface snow*
Three consecutive snow seasons were evaluated to define the chemical composition of the surface
snow in the Arctic site of GSRS. The sea salt ions Cl$^-$ (50 %), Na$^+$ (23%) represent the most abundant
species (Fig. S3), followed by SO$_4^{2-}$ (11 %), Mg (7 %), Ca (2%), Fe (1%) and Al (1%). Similar
relative abundances were also found in previous studies on the snow of the Svalbard Archipelago
(Beaudon and Moore, 2009; Vega et al., 2015; Barbaro et al., 2021; Spolaor et al., 2021b).
Table 1 reports the average ionic loads of the most abundant (> 1%) species in the surface snow,
considering three different seasons: autumn is defined until December 21$^{st}$, winter until March 21$^{st}$,
and spring from then to melt onset. The average loads of the first sampling year were lower compared
to the other campaigns (Fig. S4). The average ionic loads of the less abundant (< 1%) species are
reported instead in Table S1.

**Table 1.** Average ionic loads of the most abundant (>1%) ionic and elemental species in the surface snow during each
season of the three consecutive sampling campaigns. The standard deviation is shown in brackets, while in the case of
nss-SO$_4^{2-}$ the brackets represent the percentage of nss-SO$_4^{2-}$ compared to the total SO$_4^{2-}$. "n" indicates the number of
samples considered for the calculation of the average.

| mg m$^{-2}$ | total | Cl$^-$ | Na$^+$ | SO$_4^{2-}$ | nss-SO$_4^{2-}$ | Mg | Fe | Ca | NO$_3^-$ | K$^+$ | NH$_4^+$ |
|---|---|---|---|---|---|---|---|---|---|---|---|
| autumn 2018 (n=22) | 32 (25) | 15 (21) | 7 (11) | 3 (4) | 1 (36%) | 3 (3) | 2 (5) | 0.3 (0.3) | 0.5 (0.5) | 0.3 (0.4) | 0.04 (0.03) |
| winter 2018-19 (n=41) | 116 (80) | 55 (68) | 31 (39) | 16 (16) | 8 (51%) | 8 (8) | 0.4 (0.3) | 0.4 (0.3) | 2 (2) | 2 (2) | 0.3 (0.3) |
| spring 2019 (n=51) | 76 (50) | 36 (43) | 19 (24) | 9 (9) | 4 (48%) | 6 (5) | 2 (3) | 0.6 (0.4) | 1 (1) | 1 (1) | 0.5 (0.4) |
| autumn 2019 (n=15) | 214 (98) | 101 (71) | 40 (53) | 26 (20) | 16 (61%) | 10 (6) | 1 (1) | 9 (12) | 5 (4) | 2 (3) | 4 (5) |
| winter 2019-20 (n=43) | 339 (120) | 159 (88) | 79 (73) | 41 (20) | 21 (52%) | 16 (8) | 2 (5) | 9 (10) | 3 (2) | 3 (4) | 7 (8) |
| spring 2020 (n=49) | 273 (132) | 110 (91) | 52 (77) | 28 (25) | 15 (53%) | 21 (21) | 6 (9) | 13 (16) | 4 (2) | 2 (4) | 5 (8) |



| | | | | | | | | | | | |
|---|---|---|---|---|---|---|---|---|---|---|---|
| autumn 2020 (n=6) | 803 (542) | 435 (466) | 191 (205) | 66 (83) | 18 (27%) | 84 (165) | 1 (2) | 2 (5) | 6 (11) | 9 (10) | 2 (2) |
| winter 2020-21 (n=13) | 327 (203) | 207 (194) | 64 (48) | 41 (36) | 24 (60%) | 6 (5) | 0.2 (0.4) | 0.2 (0.1) | 5 (3) | 3 (2) | 1 (1) |
| spring 2021 (n=13) | 181 (92) | 107 (86) | 36 (26) | 16 (12) | 7 (43%) | 9 (10) | 4 (7) | 1 (2) | 3 (2) | 2 (2) | 1 (1) |

In general, the winter seasons showed the higher average loads, with the winters 2019-20 and 2020-21 being rather similar. Higher values of sea salts species were found in autumn 2020, but less snow accumulation was recorded during that period (Fig. 1).

The non-sea-salt sulfate (nss-SO$_4^{2-}$), calculated using a seawater SO$_4^{2-}$: Na$^+$ mass ratio of 0.252 (Millero et al., 2008), was the most abundant fraction of the total sulfate in autumn 2019 and winter 2020-21, while in autumn 2018 and 2020 sea salt sulfate (ss-SO$_4^{2-}$) was the dominant fraction. No clear predominance between the two fractions was achieved during the other investigated seasons (Table 1).

The abundance of all chemical species investigated is quite similar for all years (Fig. S5), although the sampling campaign 2019-20 showed higher percentage of calcium ranging between 3% and 5%, in contrast to the typical concentrations < 1% found in the other two campaigns.

*3.3 Polar vortex and Arctic Sea ice extent in 2019-20*

According to the 2023 survey conducted by the National Snow and Ice Data Center (NSIDC), the maximum extent of Arctic Sea ice since 2014 has been recorded in March 2020, with 14.73 million square kilometres of the Arctic Ocean surface, in a decadal trend characterized by a -2.53% of decline, due to the Arctic Amplification. Considering the Kongsfjorden area, the total sea ice extent varied from 63.94 km$^2$ in March 2019 to 129.81 km$^2$ in March 2020, and was with 46.26 km$^2$ lowest in March 2021 (Gerland et al., 2022). Specifications on Drift Ice (DI), Fast Ice (FI), and Open Water (OW) extent are reported in Table S2. The 2020 maximum sea ice extent followed the exceptionally strong and cold stratospheric polar vortex that took place in the Northern Hemisphere (NH) during the 2019-20 polar winter, together with low wave activity from the troposphere, which allowed the polar vortex to remain relatively undisturbed (Lawrence et al., 2020). Notably, the 2020 Arctic Sea ice extent is 16% and 9% higher than previous (2018-19) and following (2020-21) records (dataset NSIDC, NOAA), appearing more similar to Arctic type than Arctic Amplification conditions. Lower surface air temperatures, reduced precipitations, higher wind speed (m sec$^{-1}$), and minor mean snow height with respect to the typical AA conditions, were induced by strong cold polar vortex triggered by a net positive Artic Oscillation (AO) phase, and recorded in the 2019-20 winter season. The 2020 anomalous AO index is displayed in Fig. S6. Seasonal values of mean air temperatures (°C), mean





precipitation (mm), maximum mean wind speed (m sec$^{-1}$) and mean snow depth (cm) during the three
consecutive sampling campaigns are reported in Table S3. Temperature data were provided by the
Norwegian Centre for Climate Services (NCCS), while sea ice extent data were supplied by National
Snow and Ice Data Center (NSIDC). Seawater temperature data collected at 10 m depth at a mid-
fjord station near Ny-Ålesund (Kb3) was found to be colder during 2020 compared to 2019 and 2021
spring seasons (Table S4), promoting the formation of sea ice in Kongsfjorden, and supporting its
duration through the season, together with cold atmospheric conditions. Salinity data also revealed
modest fluctuations across the considered seasons, showing a decrease of 0.35 psu in 2020 relative to
2019, and a decrease of 0.1 psu compared to 2021.
**4. Discussion**
*4.1 Ny-Ålesund seasonal and interannual trends variability in surface snow*
The three consecutive sampling campaigns conducted from 2018 to 2021 confirmed the dominance
of sea salt input in the surface snow of Svalbard, likely due to the proximity of the Kongsfjord
(Barbaro et al., 2021). The dominant ions are Na$^+$, Cl$^-$, and SO$_4^{2-}$, likely associated with the
scavenging precipitation of marine aerosol (Hodgkins and Tranter, 1998). The observed mean
seasonal trends (Fig. S4) display the highest concentrations of marine species in autumn 2020,
followed by 2020-21 and 2019-20 winter seasons. However, wintry concentrations are presumably
linked to weakened (2019-20) or destroyed (2020-21) polar vortex (Fig. 1) and intense cyclonic
storms, associated with anomalous warming events capable of transporting both heat and moisture
from lower latitudes to Svalbard (Rinke et al., 2017). Autumn 2020 represents most likely an outlier,
due to scarce precipitations (Fig. 1) that led to more concentrated impurities in the surface snow.
Concerning the spring season, higher concentrations of typical marine (Na$^+$, Cl$^-$, Br-, MSA, SO$_4^{2-}$)
and geogenic (Al, Ca, Mn, Fe, Sr) species deposited in late spring 2020, compared to spring 2019
(Fig. 2), may be due to the very close drift Arctic Sea ice presence in Kongsfjorden (Table S2), which
reached its maximum extent in March 2020. Indeed, the formation of sea ice leads to the production
of highly saline frost flowers and brine at both the sea ice-ocean and sea ice-atmosphere interface.
Brine and frost flowers formed on the surface of sea ice can be lifted by winds and dispersed, thereby
increasing the concentration of sea spray aerosol in the planetary boundary layer, and subsequently
enhancing deposition over the snowpack. The maximum sea ice coverage in the fjord occurred in
March 2020 was a consequence of low-temperature anomalies and intensified atmospherically driven
sea ice transport and deformation due to higher winter wind speeds (Fig. S7), likely linked to the
exceptional occurrence of a strong and cold stratospheric polar vortex. Concurrently, an outstanding
positive phase of the Arctic Oscillation (AO) in the troposphere (Fig. 1) was recorded in January-





March 2020 (Lawrence et al., 2020; Dethloff et al., 2022), featuring as an outlier in the historical
timeseries 1950-2023 reported by the NOAA service.






















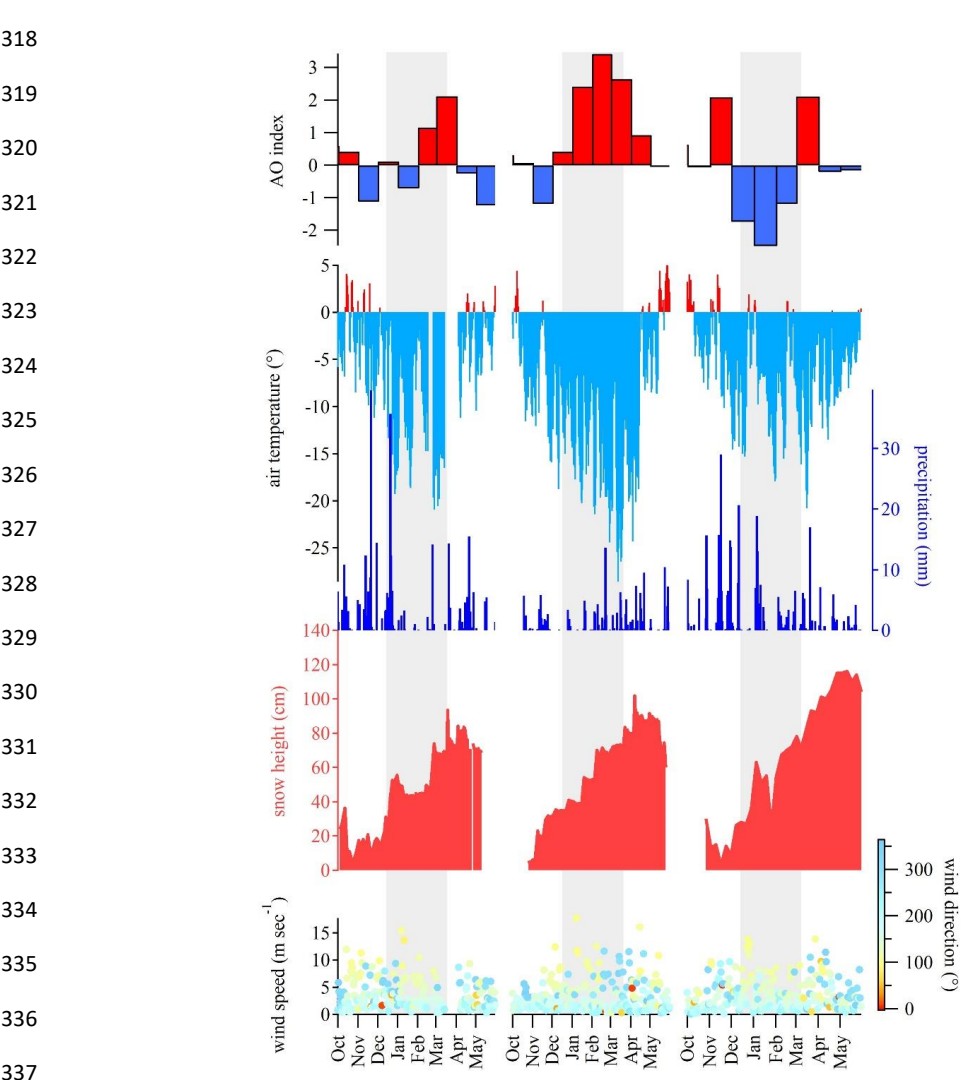

**Figure 1.** AO Index, Radiation (W m⁻²), air temperature (°C), precipitation (mm), snow height (cm), wind speed (m sec⁻
¹), and wind direction (°) from the NCEP/NCAR Reanalysis data. NOAA Physical Sciences Lab's daily composites tool
was used to calculate the near-surface air temperatures across the Northern Hemisphere from October to May.



A 2021 spring peak of marine species was also recorded, although more attenuated than spring 2020
(Fig. 1, Fig. S4). This variation is likely attributable to different extents of sea ice in the fjord.
Nonetheless, seawater temperatures in 2021, similar to those in 2020 and 2.3°C colder than in 2019
(Table S4), along with comparable wind speed conditions (Fig. S7), may also have contributed to the
observed trends in marine species concentrations. Similarly, the spring peak of Mg, Sr, Mn, Fe, Al
and V in 2021 seems to reflect the high wind speed and positive AO index recorded from March to
April 2021. In particular, positive anomalies for atmospheric (A) and wind speed (W) conditions,
together with negative oceanic (O) conditions were observed during the 2020-21 campaign, while
negative A and O conditions were accompanied to positive W during 2019-20. On the contrary, 2018-
19 diverges from the other campaigns for positive O condition associated to negative W condition
anomalies. These findings highlight the complex interplay between atmospheric patterns (AO and
wind speed), local climate (temperature and sea ice extent), and oceanic conditions (SST, salinity),
showing similar ionic and elemental trends in surface snow for wind, sea ice, and SST
counterbalanced conditions.

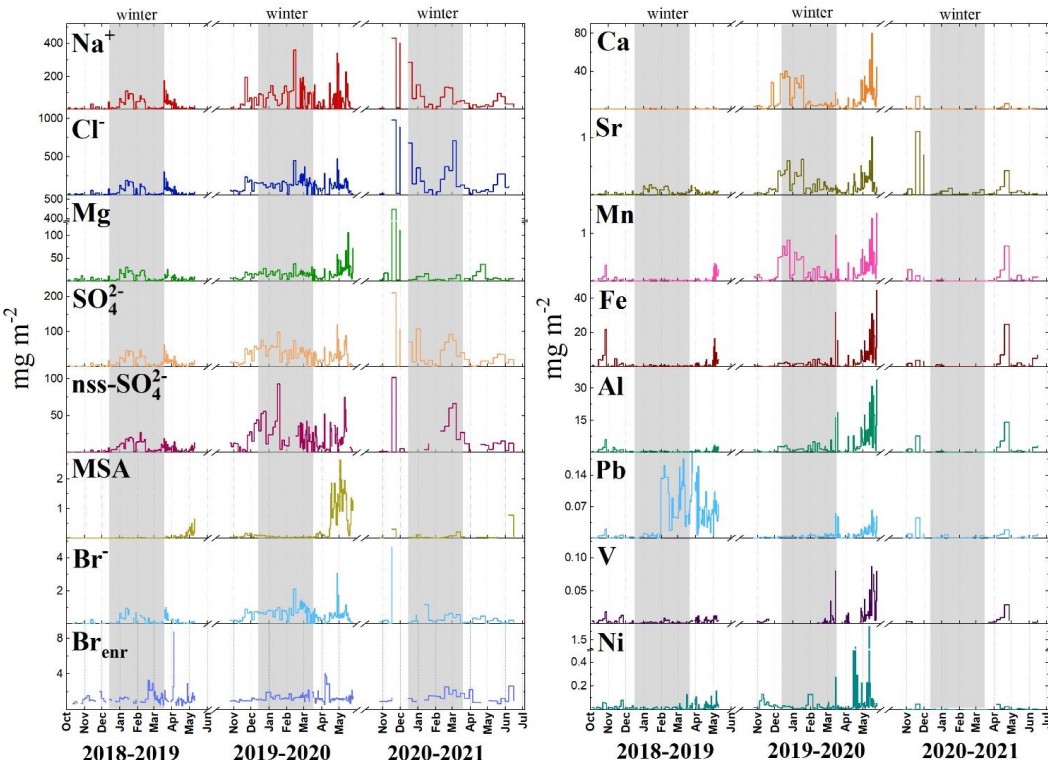


**Figure 2.** Ionic loads (mg m$^{-2}$) of Na$^+$, Cl$^-$, Mg, SO$_4^{2-}$, nss-SO$_4^{2-}$, MSA, Br$^-$, Ca, Sr, Mn, Fe, Al, Pb, V, Ni in the surface

snow for the three sampling campaigns: 2018-19, 2019-20, 2020-21.



A singular case is represented by Pb, with remarkable trend concentration revealed during spring
2019. Generally, Pb presents a typical seasonal variability in the Arctic aerosol, with higher wintry
concentration caused by seasonal differences in the mixing conditions of the troposphere (Paatero et
al., 2010). An accumulation in aerosol may lead to a prominent wet deposition in surface snow during
spring, possibly due to enhanced mixed-phase clouds' scavenging. The springtime Pb concentration
maxima are typically consistent with a mixture of eastern European, Post-Soviet States, and western
European sources (Sherrell et al., 2000; Bazzano et al., 2015, 2021). In this study, cluster mean
trajectories obtained for winter 2018-2019 highlighted a 25% of air mass provenance from Russian
Arctic and a 13% from eastern Siberia (Fig. S8), possibly explaining the higher concentrations of Pb
revealed in spring 2019, following a reduced precipitation regime that occurred in January 2019. A
local anthropogenic origin can be excluded though, since no activities were recorded in the vicinity
of the sampling site in 2019. In addition, both GSRS and Ny-Ålesund (Fig. S2), located at 1 km of
distance from each other, recorded comparable high concentrations of Pb, thus ruling out a possible
contamination. However, at present, the long-range transport of Pb remains a hypothesis, likely
supported by the breakdown of the wintry polar vortex (Fig. 1). To clarify the origins of Pb peaks
recorded between winter and spring 2019 further investigations are needed, which goes beyond the
scope of this study.
Other backward trajectories (Fig. S8) for Ny-Ålesund area (78.92° N, 11.89° E) appear mostly in line
with literature findings (Platt et al., 2022; Vecchiato et al., 2024), showing three main seasonal
characters: a prevalent mass movement from ice-covered Central Arctic Ocean, Kara Sea, and
Greenland Sea during autumn, a main provenance from Central Arctic Ocean and Kara Sea during
winter, and a predominant trajectory from Northern Canada in addition to air masses arriving from
Arctic Ocean and Kara seas during spring.

382        *4.2  The main ion sources in the seasonal snow of Ny-Ålesund*

Looking at the dominant ions associated to the marine aerosol, we found $Cl^-/Na^+$ median ratios
ranging from 1.3 to 1.5 w w$^{-1}$, slightly lower than the expected value of 1.8 w w$^{-1}$ in the pure seawater
(Zhuang et al., 1999), pointing the occurrence of a minimum $Cl^-$ depletion in aerosol, quantified as
14% for the 2018-19 and 2019-20 campaigns, and as just 2% for the 2020-21 campaign. A possible
explanation for this phenomenon could be the de-chlorination of sea-spray aerosol during transport,
or, less likely, at the snow-atmosphere interface; while a possible influence of biomass burning on
$Cl^-$ depletion process has been excluded by the negative correlation found between $Cl^-$ depletion



values and nss-K$^+$/K$^+$ ratios, which is a tracer of relative contribution of biomass burning (Song et al.,
391  2018).

Mg, Ca, and K$^+$ appear positively correlated with Na$^+$ and Cl$^-$, which may indicate a common sea-
spray source. However, the concentrations of Mg are also positively correlated with nss-Ca ($\rho_{load}$ =
0.55), suggesting that they share some non-marine source(s). Moreover, surface snow samples
collected during the three campaigns had greater Ca : Mg ratios than seawater (0.32, Millero et al.,
2008), pointing that the excess of these ions may come from mineral particles (i.e., calcite and
dolomite), derived from local rock or soil dust (e.g., limestone, dolostone, and marble, which are
abundant in Svalbard), as previously observed by Barbaro et al. (2021).
Additionally, sulfate (SO$_4^{2-}$) is highly and significantly correlated (p < 0.05) with both Na$^+$ ($\rho_{load}$ =
0.76) and Cl$^-$ ($\rho_{load}$ = 0.93), indicating that sea-spray is its main source. Nonetheless, Na$^+$/SO$_4^{2-}$ and
Cl$^-$/SO$_4^{2-}$ ratios are significantly lower than typical seawater values (3.97 and 7.13, respectively,
according to Millero et al., 2008) for the former two campaigns (2018-19, 2019-20). This indicates
an input of nss-SO$_4^{2-}$, which may originate from crustal inputs, the transport of anthropogenic
compounds (e.g., emissions from fossil fuels), or by the oxidation of dimethylsulfide (DMS) released
from marine biological activities. To quantify the biogenic nss-SO$_4^{2-}$ contribution, the
methanesulfonic acid (MSA) loads - the final product of DMS oxidation - were multiplied by 3.0
(Udisti et al., 2016), revealing biogenic SO$_4^{2-}$ contributions ranging from 0.15% (2018-19, 2020-21)
up to 0.38% (2019-20). Furthermore, the MSA/nss-SO$_4^{2-}$ ratio was inspected, revealing a mean value
of 0.02 ± 0.03 during the first (2018-19) and the third (2020-21) sampling campaigns, and a maximum
ratio equal to 0.06 ± 0.18 reached during the second campaign (2020-21), similar to the subarctic
western North Pacific ratio found by Jung et al. (2014). However, several factors can influence MSA
formation, a univocal marker of biogenic emissions, including higher biological productivity related
to higher nutrient input; the concentrations of NO$_3$ radicals as key oxidants for DMS decomposition
(higher NO$_3$ gives higher MSA); and lower air temperatures, which tend to yield higher MSA levels
(Andreae et al., 1985; Udisti et al., 2020). For the 2019-20 campaign, it seems likely that a
combination of these three factors, together with the positive expansion of sea ice and the very close
drift ice presence in March 2020, as revealed from satellite reconstructions (Fig. S9), contributed to
the increased release of MSA in aerosol, and its consistent deposition in surface snow (Fig. 2). Indeed,
DMS was likely accumulated under the sea ice cover in the fjord and surrounding areas, and then
being released and oxidised in atmosphere when the ice broke off and melted (April-May).
Furthermore, lower temperatures, highly positive correlation between MSA and NO$_3^-$ ($\rho_{load}$ = 0.64),
and short-range transport from the source to the near-coast sink site (GSRS) would have aided



elevated concentrations of MSA in atmospheric depositions. Contrarily, in the 2018-19 season, the
sea ice melted significantly earlier, possibly not allowing enough time with adequate sunlight for
substantial biological activity to accumulate beneath or within it. This occurred despite the dominance
of a species known for high DMS production in 2019, unlike the following year, according to Assmy
et al. (2023).
The crustal fraction of sulfate (cr-$SO_4^{2-}$) was estimated by multiplying the nss-Ca (as crustal marker)
content by 0.59 ($SO_4^{2-}$/Ca w/w ratio in the uppermost Earth crust - Wagenbach et al. 1996), obtaining
variable contributions for the three sampling campaigns, ranging from 2.45% up to 12.94%.
The anthropogenic contribution to nss-$SO_4^{2-}$ concentrations was also investigated by the application
of the [ex- $SO_4^{2-}$] concentration formula, considering the average concentration of [Ca] instead of the
average ionic concentration [$Ca^{2+}$] for the already clarified reason:
[ex- $SO_4^{2-}$] = [$SO_4^{2-}$] – (0.12 [$Na^+$]) – (0.175 [$Ca^{2+}$])
The obtained results showed a 50 up to 60% of anthropogenic contribution for the nss-$SO_4^{2-}$ input,
corroborating previous results showed for the same area by Amore et al. (2022). The plausible source
of the anthropogenic fraction is the atmospheric transport of secondary aerosols containing $SO_4^{2-}$, and
ammonium sulfate. This sulfate can be formed by $SO_x$ emitted from coal combustion throughout the
winter and biomass burning in the spring (Barbaro et al., 2021 and reference therein). The nss-$SO_4^{2-}$
does not correlate significantly with other ionic species (except for Mg), thus suggesting a separate
origin.
The ammonium ($NH_4^+$) load showed significant positive correlations with $Na^+$ ($\rho_{load} = 0.76$), $Cl^-$ ($\rho_{load}$
$= 0.62$) and $K^+$ ($\rho_{load} = 0.75$), as well as with $SO_4^{2-}$ ($\rho_{load} = 0.62$), $NO_3^-$ ($\rho_{load} = 0.58$), MSA ($\rho_{load} =$
$0.52$) and $Br^-$ ($\rho_{load} = 0.62$), suggesting a close link with sea-salt ions and biogenic emissions, rather
than anthropogenic activities, although some contribution from biomass burning events cannot be
excluded.
*4.3 Bromine enrichment*
The bromine enrichment factor ($Br_{enr}$) is well known to reflect specific processes (i.e., sea ice gas
phase $Br^-$ emission) that affect the $Br^-$ concentration and load in the snowpack (Spolaor et al., 2014).
Therefore, calculating the relative enrichment over the Br/Na ratio in sea water can offer crucial
insights on sea ice variability for the investigated Arctic areas (Barbaro et al., 2021). As reported in



previous studies (Maffezzoli et al., 2017; Barbaro et al., 2021), the Br enrichment factor ($Br_{enr}$) can
be calculated as $Br_{enr} = Br^- / (0.0065\ Na^+)$, where 0.0065 represents the $Br^-$ : $Na^+$ seawater mass ratio.
On the contrary to what observed in a former study (Barbaro et al., 2021) for the Hornsund area and
north-western Spitsbergen, where the $Br_{enr}$ mean values were often < 1, indicating some $Br^-$ depletion,
in this study we observed $Br_{enr}$ mean values ranging from 1.5 up to 17.7, with the highest value
associated to the second sampling campaign conducted in 2019-20, which showed the most extensive
sea ice coverage. These results support the impact of the sea ice expansion and the close drift ice in
the Kongsfjorden on the snow chemical composition. Indeed, newly formed sea ice releases gas-
phase $Br^-$ into the polar atmosphere, thus supplying an extra $Br^-$ source in addition to sea spray
(Spolaor et al., 2016).
*4.4 Anthropogenic and natural sources of ions and particulate trace elements*
To distinguish possible anthropogenic contributions from natural ones (marine and geogenic) for ions
and particulate trace elements, a Hierarchical Cluster Analysis (HCA) method was carried out.
Results of clustering (Fig. 3) clearly disentangle marine ($Na^+$, $Cl^-$, $K^+$, $NH_4^+$, $SO_4^{2-}$, $NO_3^-$, $Br^-$),
anthropogenic (Mg, Ba, Bi, Cr, As, Ag, Cd, Pb, Cu, Ni), and geogenic (Al, Cs, Co, Rb, Fe, Be, Se,
Ca, Mn, Li, Sr) sources of ionic and elemental species. Interestingly, biogenic MSA is brought
together with the anthropogenic cluster, likely due to the coincidence of an algal bloom event with
the major deposition of anthropogenic metals in surface snow. Although winter is the most eligible
season for greater deposition of impurities due to favorable atmospheric conditions, Pb, and Ni show
higher concentrations in spring 2019 and spring 2020, respectively (Fig. 2), representing the indicator
of anomalous atmospheric and depositional events. However, in the absence of detailed information
on the size of the particles, and on the isotopic composition of the investigated elements, which may
distinguish local from long-range transport pollutants, no definitive statements can be made about the
sources of these impurities.






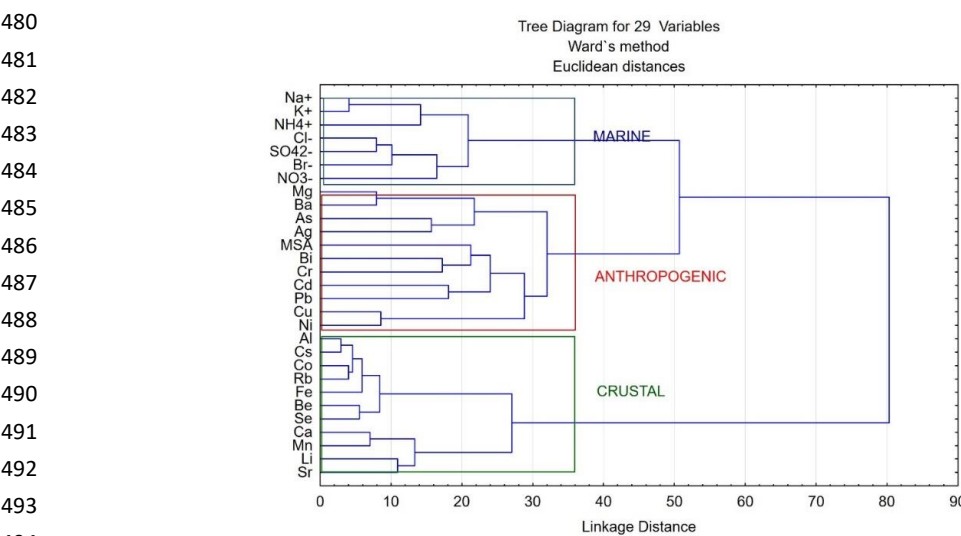

**Figure 3.** Hierarchical cluster analysis applied to further disentangle the particulate trace element non-crustal sources.

## 5. Summary and Conclusion

In this study, trace elements and major ions were investigated in surface snow samples collected in Ny-Ålesund between October 2018 to June 2021. Seasonal and interannual variations of impurities have been observed, with general higher concentrations of marine species revealed in late spring 2020, associated to Arctic type conditions, and attributed to more extensive sea ice in Kongsfjorden in March 2020, promoted by negative temperature anomalies in both atmosphere and ocean and likely related to higher air mass recycle within the Arctic. In fact, sea ice has a role in concentrating, storing, and releasing marine species, as well as influencing atmospheric and oceanic processes that affect their production and distribution. Higher concentrations in spring 2020 for geogenic and anthropogenic species were attributed instead to higher wind speeds, low atmospheric temperature anomalies, and generally drier conditions resulting from the exceptional occurrence of a strong and cold wintry stratospheric polar vortex, accompanied by an unprecedently positive phase of the Arctic Oscillation in the troposphere during January-March 2020. Therefore, our results highlighted a close dependence of high concentrations of impurities found in the snowpack at Ny-Ålesund on meteorological conditions, especially during cold years, when the production of sea spray related aerosol likely derives by a larger extension of sea ice and stronger local Arctic circulation. From the comparison with previous and following seasons, the 2020-21 and 2018-19 were recognised as typical years of Arctic Amplification conditions, whilst the 2019-20 sampling campaign year has been assimilated to the Arctic type conditions. Furthermore, the identification of geogenic, marine, and





anthropogenic sources in the snowpack was allowed by a chemometric approach (HCA), which
brought to light an unexpected positive correlation between MSA and anthropogenic impurities
during the 2020 spring season. This relation can likely be attributable to the coincidence of early
spring algal bloom events with the major deposition of anthropogenic derived elements in surface
snow consequent to a wintry retention of these pollutants in the atmosphere, due to a former reduced
precipitation regime. Finally, back trajectories were realized, and three seasonal features were
identified, with a prevalent air mass provenance from circumpolar Arctic area during fall and winter,
and a predominant trajectory from Northern Canada in addition to air masses arriving from Arctic
Ocean and Kara seas during spring. On the contrary, no prevalent mid-latitude air currents were
revealed in spring as expected, considering the period of the three sampling campaigns (2018-2021).
Our results highlight the complex interplay between atmospheric patterns, local and oceanic
conditions that jointly drive snowpack impurity amounts and composition.

**Data availability**

The data supporting the findings of this study are available within the article and its supplementary
materials. Other data that support the findings of this study are available from the corresponding
author upon request.

**Author contribution**

AS: Conceptualization, Data curation, Investigation, Writing-original draft, Writing-review and
editing. EB: Conceptualization, Field work, Data curation, Formal Analysis, Writing-original draft,
Writing-review and editing, Funding acquisition. MF: Formal Analysis, Field work, Data curation,
Writing-review and editing. FS: Field work, Formal analysis, Investigation, Writing-review and
editing. MV: Writing-review and editing. MV: Field work, Writing-review and editing. MM:
Investigation. FB: Investigation, Writing-review and editing. FB: Investigation. Field work. CJMH:
Investigation, Data curation, Writing-review and editing. AB: Field work, Data curation, Writing-
review and editing. AG: Resources, Supervision, Validation, Writing-review and editing, Funding
acquisition. CB: Resources, Supervision, Validation, Writing-review and editing, Funding
acquisition. AS: Funding acquisition, Supervision, Validation, Writing-review and editing.

**Competing interests**

The authors declare that they have no conflict of interest.



**Acknowledgments**

This project has received funding from the European Union's Horizon 2020 research and innovation programme under grant agreement no. 689443 via ERA_PLANET Strand 4 project iCUPE (Integrative and Comprehensive Understanding on Polar Environments). We are grateful to the Arctic Station Dirigibile Italia from the Italian National Research Council – Institute of Polar Science (CNR-ISP) for logistical support. The authors gratefully acknowledge Claudio Artoni, Maria Papale, Ivan Sartorato, Federico Dallo, Alice Callegaro, Marco Casula, Mariasilvia Giamberini, Fabio Giardi, and all the station leaders of "Dirigibile Italia" who participated and offered valuable help and logistic support during the 2018-2021 sampling campaigns. Furthermore, we acknowledge Klara Wolf, Linda Rehder, and Ane Kvernvik for help with CTD sampling. We acknowledge the help of ELGA LabWater in providing the PURELAB Pulse and PURELAB Flex, which produced the ultrapure water used in these experiments.

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
