# Peer review of "Svalbard surface snowpack Azzurra Spagnesia,b, Elena Barbaroa,b \*, Matteo Feltraccoa,b, Federico Scotoc,b, Marco Vecchiatob, Massimiliano Vardèa, Mauro Mazzolaa, François Burgayd,e, Federica Bruschif, Clara Jule Marie Hoppeg, Allison Baileyg, Andrea Gambaroa,b, Carlo Barbantea,b, Andrea Spolaora,b a Institute of Polar Sciences - National Research Council of Italy (ISP-CNR), Via Torino 155, 30172,"

_EGUsphere, 2024_

## Author Response (AR1)

**REFEREE 1**

Review of Impact of Arctic Amplification variability on the chemical composition of the snowpack in Svalbard (Spagnesi et al., 2024).

**General comments**

The work of Spagnesi et al. discusses the temporal (and, to a lesser extent, spatial) variability of ionic and elemental concentrations in surface snow samples collected from Ny Ålesund and GSRS (Svalbard) at variable frequency during the Arctic "snow season" (i.e., autumn, winter and spring) of three consecutive years. The authors compare the snow chemical budget of years 2018-2019, 2019-2020, and 2020-2021, separating autumns, winters and springs. The authors interpret the ions and trace elements temporal trends in the light of climatological observation data (air temperature, wind speed and direction, precipitation, sea water temperature, sea ice extent), and the arctic oscillation (AO) index. The source apportionment of the various chemical species measured in the snow is supported by statistical analyses (correlations, multivariate analysis, ratios) and by modelled backward atmospheric trajectories.

Sustained high concentrations in sea salt ions during winter and spring 2020 are attributed to the particularly cold and windy conditions in the Arctic, and to the presence of sea ice in Kongsfjorden late spring that year. These conditions were favored by a strong polar vortex due to a pronounced positive AO index situation, which developed in January 2020. The reduced sea salt content in the winter-spring snowpack of years 2019 and 2020 is interpreted as a characteristic of warm years when the Arctic Amplification phenomenon is at play. Therefore, the authors conclude that the chemistry of Svalbard snowpack is directly impacted by the Arctic Amplification variability (which is a bit awkward because the AA is not a defined/quantifiable index but rather a collection of multiples feedback. In that sense, AO would be a more tangible reference).

This study presents a rare dataset of surface snow chemistry at high Arctic site that is advantageously well monitored all year round for various atmospheric parameters (including aerosol composition and climatological parameters). This work is therefore a welcome contribution in the understanding of aerosol emission over the sea ice. It also contributes to the determination of whether the amount of sea salt deposited at a coastal Arctic site is controlled by sea ice or by meteorology (wind direction and speed, aerosol transport and deposition). Resolving this question is, among other things, crucial in the interpretation of sea salt records from Svalbard ice cores (Rhodes et al., 2018). Unfortunately, the disorganized structure of the manuscript does not do justice to these valuable data. The results and discussion jump back and forth between addressing the interannual variability, the seasonal variability, the 3-year average, and between explaining the concentration trends and the source apportionment of many chemical species. All these dimensions of the discussion could be re-organized and more focused to facilitate its understanding and clarify its logic.

I found the authors are sometimes very specific and sometimes rather vague in their explanations of processes involved in the production of the various types of aerosols.

On the more scientific side, I have some reservations about the author's interpretation of the trace elements trends (especially Pb) and source apportionment (see below).

For these reasons, I can only recommend the publication of this manuscript if major revisions are undertaken.

Let me point out some specific issues and suggest a way to resolve them in the "specific comments" below. Concerning the discussion/interpretation points that were not clear to me, I am, at times, just asking a question to the authors, which I am hoping they would answer in the discussion, and preferably address in their revised manuscript.

**Specific comments**

The abstract, introduction and methodology sections are well detailed and well written. I have a couple of questions about the surface snow sampling:

- R-1: How was the spatial variability (within the 100m wide snowfield) taken into account? Aerosols (and snow) are not deposited uniformly on the surface. Since the samples are so small and because samples are taken every 3-5 days, how is spatial variability integrated in the overall sampling plan? (i.e., the ion concentrations can vary as much between adjacent samples as it varies between samples taken at the same spot but on two consecutive day). In other words, what is the magnitude of spatial variability of ion concentrations vs the magnitude of the temporal variability?
  - A-1 We thank the referee for raising this important point, as spatial variability can indeed have a significant impact on the results presented. First, it is necessary to mention that while the snowfield considered a "clean area" for sampling is 100 meters wide, the actual snow sampling area is much smaller, around 3x3 meters, and is located near the nivometric station. A similar sampling strategy has been used in previous studies, and spatial variability was evaluated in these cases. For example, in Spolaor et al. (2019), spatial variability was tested as follows: At the beginning of the experiment, six samples were collected to assess the spatial variability of mercury, iodine, bromine, and sodium in surface snow within the delimited snowfield. The spatial variability was analyzed across three experiments, specifically for these four elements. The six surface snow samples were collected simultaneously within the defined area at the start of each experiment. The results indicated that spatial variability accounted for approximately 10% of the variability for sodium, bromine, and mercury, while for iodine, the variability was around 5%. Similarly, in Bertò et al. (2021) the spatial variability for rBC (refractory Black Carbon) in the same snowfield was evaluated: The spatial variability in rBC, calculated in the same manner as proposed by Spolaor et al. (2019) for other species, was determined from six surface snow samples taken from the four corners and two from the center of the sampling area before the experiment. The rBC mass concentrations obtained were as follows: (a) 3.95 ng/g, (b) 4.92 ng/g, (c) 4.20 ng/g, (d) 3.10 ng/g, (e) 3.82 ng/g, and (f) 3.58 ng/g. This resulted in an rBC spatial variability of 16% in the surface snow of the sampling area. In addition, we tested the surface concentrations (upper 3 cm) of the investigated ions in a snowfield near Ny-Ålesund and in Gruvebadet, approximately 800 meters apart. The results showed very similar concentrations. The previous assessments of spatial variability, combined with the results from the two simultaneous sampling sites, suggest that while spatial variability may affect the signal, it likely remains within a 5-15% range.
- R-2 The authors could include a schematic of the snow sampling layout.
  - A-2 We do not believe it is necessary to include a detailed sampling scheme, but a brief description has been added to the manuscript (lines 137-142). The same protocol used in Spolaor et al. (2019) was followed for sample collection (with a lower temporal resolution), where consecutive and adjacent sampling was employed. Each sample was collected no more than 10 cm apart from the previous one along a precise path and within the delimited snow field. This method was designed to minimize the variability between consecutive samples and reduce the impact of potential spatial variability, ensuring a more consistent dataset. For additional information on the snow sampling area please also look the paper Scoto et al. 2023 (https://doi.org/10.3389/feart.2023.1123981)
- R-3 If surface snow is sampled 3 days apart and it has not snowed during these 3 days, does it mean that the two samples represent the same snow layer, except that after 3 days the snow has aged, and therefore the chemistry changed?
  - A-3 Correct, in the absence of snowfall (or wind drift/erosion), the two samples represent the same snow layer. However, this does not necessarily imply that their concentrations will be identical. Factors

such as snow aging, potential element re-emission and transformation, or dry deposition can introduce variability. Additionally, considering that the sampling site is 1 km from the shore, marine emissions are a significant factor, as are atmospheric transport and dry deposition processes, all of which can influence the elemental concentrations in the snowpack.

- R-4 How is it possible to know when a snow layer has been ablated or eroded by wind or drift? A-4 The snow ablation and erosion can be inferred from snow height measurements and the Δh (change in snow height) between consecutive sampling events. A decrease in snow height indicates ablation, which may result from snow melting (in cases of positive temperatures), snow aging, sublimation, and/or snow drift. In general, a wind speed of 5 m/s is considered the threshold for initiating snow drift and ablation episodes (Pomeroy, 1989). We included ΔH discussion in the manuscript and in Figure 1, considering instances where snow ablation coincides with wind speeds equal to or greater than 5 m/s.
- R-5 I suggest the concentration time series be plotted along with the precipitation data and wind direction. That way the reader can directly match fresh precipitation and sea salt (or crustal) input.
   A-5 We agree with this suggestion and plotted the precipitation data and wind direction against the concentration ts.
- R-6 As a general comment for the whole structure of the paper, let me suggest the authors to begin with determining the sea-salt, non-sea-salt, crustal, and biogenic proportions of the ions. Proceed in order to avoid jumping back and forth among various ions. For example, in section 4.2., Cl is discussed, then Mg, then SO4, then MSA, then SO4 again, NH4. Then Br- has a section of its own. Once we know precisely what is the nss faction of, let's say, SO4, let's stick to it and use it in other calculations and plot is in the figures. As it is now in the manuscript, different parameters are invoked in different interpretations, which is confusing.
  - A-6: We appreciate the suggestion to reorganize the text to first determine the sea-salt, non-sea-salt, crustal, and biogenic fractions of the ions. In response, we have made targeted revisions to Section 4.2 to reduce jumps among different ions. However, we would like to explain that the original structure was intentionally designed to highlight the diverse sources and their relative contributions for each ion, given that many originate from multiple sources. We are concerned that a rigid classification might lead to artificial distinctions that do not accurately represent the complexity of the data. We emphasize that, although we may not always be able to distinctly isolate sources, our discussion reflects the most likely origins for each case.
- R-7 There are too many figures in the supplementary material. Many of which could be removed because they do not bring much to the discussion or because they are redundant (e.g., S3, S4, S5).
   A-7 Thanks for the valuable feedback. Following the referee's suggestion, we removed the redundant Fig. S3 and S5, while retaining Fig. S4, which highlights the concentration distributions across different seasons.
- R-8 In section 3.1. Ca2+ is used as a crustal tracer, why not nss-Ca? Also, how does Ca compares with other crustal elements such as Fe or Al or Ba? In Fig. 2, it seems that the peak observed in April May 2020 could be a crustal input.
  - A-8 At this step, we used Ca2+ rather than nss-Ca as a crustal tracer to prioritise the direct measurement of Ca2+ and avoid the potential errors introduced by nss-Ca calculations. Additionally, the enrichment factor (EF) we calculated, using Ba as a conservative reference element (Wedepohl, 1995), consistently showed EF values below 10 throughout the 2018-19 sampling season (Oct-May

2018-19, represented in Fig. S2, section 3.1). This low EFs confirm a predominantly crustal origin for Ca2+, supporting its reliability as a tracer in this context.

We acknowledge the referee's suggestion regarding the comparison of Ca with other crustal elements such as Fe and Al. We agree that the peak observed in April-May 2020 is suggestive of a crustal input.

• R-9 My suggestion here is to calculate the crustal enrichment factor of all trace elements using either Fe or Al as a crustal element of reference. Any significant enrichment would mean that the source of the element of interest is not crustal but is likely anthropogenic. The enrichment factor (as described for instance in Ruppel et al., 2023) should then be used in the cluster analysis instead of the concentrations. Also, the cluster analysis should include nss-SO4, nss-Ca, biogenic-SO4, which would be more insightful that the total fraction of these ions. It is a pity to have made the effort to calculate the various contributions to the ionic budget and then not use these calculated parameters in the cluster analysis nor in the time series plots.

A-9: We thank the referee for highlighting this point and we computed the Enrichment crustal factors of all trace elements, however choosing Ba as a conservative crustal element (Widepohl, 1995), for its solubility, abundance, and the negligible effect of sea spray emissions on its concentration. We integrated the results in the manuscript, to implement the discussion and disentangle the probable sources of TEs. As suggested by referee, we performed another cluster analysis including nss-SO4, nss-Ca, biogenic-SO4, obtained a more specific results for each species. We prefer to not include the enrichment factors in the cluster analysis. Enrichment factor is an indirect measurement because it is calculated by comparing the concentration of a specific trace element in the sample to its concentration in a reference material. The uncertainty on the reference material, not specific for the Svalbard area, can improve the uncertainty on the results of cluster analysis where real analyses of the samples are included.

- R-10 For Pb, it may turn out that the 2019 peak is enriched (supporting the anthropogenic source) and the 2020 peak is no longer a peak in Pb enrichment factor (snow not enriched in Pb compared to the continental crust), which would suggest the 2020 April-May peak is mostly crustal (dust peak due to strong wind? Or early snow melts and exposed surrounding? Etc.)
  - A-10: We totally agree with the referee comment, since EFs calculations support this interpretation. Therefore, we revised the Pb part in the manuscript on the light of this observations (L 524-528).
- R-11 Coming back to the sea salt peak in winter 2020: have the authors tried to detect the contribution of frost flowers formed on young sea ice? Several authors (e.g., Rankin et al., 2002; Beaudon and Moore, 2009, Roscoe et al., 2011) verify the influence of mirabilite deposition in the snow simply by calculating nss-SO4 and the SO4/Na ratio. Perhaps the authors could try a similar analysis? This could potentially support the direct impact of sea ice in Kongsfjorden on the land snowpack chemistry. A-11 We greatly appreciate the reviewer's suggestion, and we did attempt to detect the contribution of frost flowers by calculating both the SO4/Na ratio and nss-SO4 as suggested. Our results showed that for all the investigated seasons, including the winter 2020 peak, the SO4/Na ratios were consistent with that of seawater (0.25). Notably, winter 2019-20 displayed a slightly lower median ratio (0.21), compared to the other inspected winters (0.31 for winter 2018-19 and 0.30 for winter 2020-21). This slight deviation suggests a reduced relative contribution of sulfate compared to sodium in 2019-20, but not to a degree that would indicate substantial mirabilite precipitation or a clear frost flower influence.

While the consistency of the SO4/Na ratio with seawater points to a dominant marine aerosol source, the slight decrease in the 2019-20 season could reflect specific meteorological conditions, such as changes in sea ice coverage or atmospheric transport patterns. We also examined nss-SO4 and found no significant depletion in snow, further supporting that mirabilite deposition likely played a minimal

role. Therefore, while frost flowers are known to influence snow chemistry in polar regions, our analysis suggests that their contribution to the observed sea salt peak in winter 2020 was limited in Kongsfjorden.

**Technical corrections**

Here are some edits to be considered:

- R-12 I. 90: replace "far" by "south"
   A-12 We replaced "far" by "south" as suggested
- R-13 Fig. S1: the Svalbard map insert is too small.
   A-13 We agree with the referee and decided to substitute the Svalbard map.
- R-14 l.235: "depth", do you mean "thickness"?
   A-14 Yes. We replaced "depth" by "thickness"
- R-15 Fig. S3: was Mg2+ measured by ion chromatography? How does it compare with total Mg measured by ICP-MS?
  - A-15 Yes, we measured both Mg2+ and total Mg (as well as Ca2+ and total Ca) using ion chromatography (IC) and ICP-MS, respectively. The Mg2+ concentrations were generally lower than total Mg (except for winter 2020-21), reflecting the soluble fraction compared to the total elemental content. Notably, Mg2+ represented around 1.25%, compared to 3.02% for total Mg, which is roughly half, if considering the first campaign only.
  - However, Mg2+ and Ca2+ were not measured in samples from the second campaign (2019-20) due to instrumental limitations, therefore we used the ICP data that we have available for all campaigns.
- R-16 Table 1 and Table S1: A box plot (similar to that of Fig. S6) would be easier to read than these tables
  - A-16 We thank the referee for this suggestion. We explored the option of presenting the data as box plots instead of tables, but found that it did not improve readability. Given the need to display 33 variables (trace elements and ions) across 9 seasons, the box plots either showed each analyte without clearly distinguishing the seasons or grouped the seasons with a summary of all ions and elements, making the visualization difficult to interpret. Therefore, the tables provided a more practical and clear representation of the data.
- R-17 l. 248: "average", why not using medians? Standard deviations are so large.
  - A-17 We appreciate the referee's suggestion regarding the use of medians rather than averages, especially given the large standard deviations observed in some cases. We opted to present averages because our primary goal was to reflect the overall ionic and elemental loads across the different seasons, including the influence of occasional high-concentration events, which can be important for understanding episodic phenomena or extreme environmental inputs. Averages, while sensitive to such high concentrations, provide a more complete picture of the total ionic and elemental deposition during the campaign periods. While medians are less influenced by outliers, they may underrepresent the significance of such events, which we consider critical for understanding the snowpack chemistry dynamics in this Arctic environment. That said, the large standard deviations reported alongside the averages indicate the variability and outliers within the data, ensuring transparency about the spread in concentrations. To address the referee's concern,

we can include the median values in supplementary materials or provide additional commentary in the text to clarify how episodic events may have influenced the overall averages.

• R-18 l.255-256: "less snow accumulation", could there be dry deposition?

A-18 In order to clarify if the dry deposition may have had a role, in the higher values found in autumn 2020, we computed Sperman rank correlations ( $\rho_{load}$ ) between the total ionic loads to examine the relationships between different species and their deposition modes (Barbaro et al., 2021). We did not find any positive correlations between sea-salt ions (Na+, K+, NH4+, Cl-, SO42-, Br-, NO3-) and non-marine ions (Ca2+, Mg2+) during Autumn 2020, suggesting that mainly wet deposition contributed to the snowpack, during that season. Therefore, the less snow accumulation hypothesis is more probable than dry deposition.

• R-19 l.262: could the higher percentage of calcium in 2019-2020 invoked here be related to a higher crustal contribution (dust).

A-19 Yes, but except for autumn 2019, when Ca EF values are greater than 10, indicating a mixed source.

- R-20 Fig. S6: please write the median values below the median bar so that the number is readable.
   A-20 We wrote the median values within the boxes, below the median bar, to made the number readable.
- R-21 Table S3: please indicate what is the location for temperature measurements. A-21 We added the location (Ny Ålesund) to Table S3.
- R-22 Fig. S7: what are the blue rectangles?

A-22 We added an explanation for blue rectangles in Fig. S5, ex Fig. s7 They highlight the recorded wind speed values during the observed spring seasons.

- R-23 Fig.1: please add the years under the x axis. What are the grey bands?
   A-23 We added the years as suggested. The grey bands indicate the winter periods, as highlighted in the caption.
- R-24 l.343: "Fig. 1" do you mean Fig.2?
   A-24 Yes, thanks for observing it. We changed to Fig. 2.
- R-25 l.359 to 375: This entire paragraph discussed the source of lead but concludes by saying the origin of Pb peaks is beyond the scope of this study. Please chose to either present and discuss lead or to ignore it. I believe that once the Pb enrichment factor will be calculated, the discussion of weather Pb is of anthropogenic origin or of crustal origin will be easier. This paragraph is the weakest of the whole paper, as it seems overinterpreted.

A-25 We thank the referee for highlighted this point, and, as suggested, we computed the crustal EFs, then discussed the origin of Pb peaks in light of the obtained results.

• R-26 l.361: Paatero et al., 2010 discussed 210Pb, which is not directly comparable to 208Pb that was measured by the authors.

A-26 We agree with the referee's observation and apologize for the oversight. To avoid any confusion, we have decided to remove this sentence from the manuscript.

- R-27 L.362: What does this sentence mean?
  - A-27 The intent of this sentence was to explain that during winter, aerosols (including pollutants like lead) accumulate in the Arctic atmosphere. With the arrival of spring, precipitation (rain or snow) removes these particles from the air, depositing them onto the snowpack. Mixed-phase clouds, which are particularly efficient at capturing aerosols, enhance this process by 'scavenging' the particles and contributing to increased deposition. However, we have decided to remove this sentence as it may sound too general and instead focus on the findings from the enrichment factor (EF) analysis.
- R-28 l.367-368: What proves that Pb coming from Russia is of anthropogenic origin? If it were from anthropogenic origin, what process would be involved? Mining? Smelting? Fossil fuel combustion? Other industrial process? Why no other anthropogenic trace metals peak at the same time as Pb? A-28 The computed enrichment factors (EFs) for lead (Pb) indicated values greater than 100, which strongly suggest a significant anthropogenic contribution. The observed springtime maxima in Pb concentrations are consistent with a mixture of emissions from eastern and western European sources. These regions have historically been associated with industrial activities that release Pb into the atmosphere. Potential processes contributing to the anthropogenic Pb observed in our samples could include mining and smelting operations, as these activities are known to release heavy metals into the environment. Furthermore, fossil fuel combustion—especially from industrial facilities and power plants—represents another significant source of atmospheric Pb. While we cannot definitively attribute the observed Pb to a specific source without isotopic analysis, the historical context suggests that these processes are likely contributors. Regarding the absence of concurrent peaks for other anthropogenic trace metals, it is important to note that Pb has unique emission profiles compared to other metals. Factors such as specific industrial practices, regulatory changes, and the varying use of leaded fuels can lead to discrepancies in the timing and concentrations of different metals. Additionally, weather patterns and atmospheric transport mechanisms can influence how and when these metals are deposited in the Kongsfjorden area. To further investigate the sources of Pb, additional studies involving isotopic characterization and particle size analysis would be necessary. These analyses could provide deeper insights into the origin and transport pathways of Pb in the region, enabling us to better understand its anthropogenic nature.
- R-29 I.369: "no activities were recorded", which activities? What kind?
   A-29 Ny-Ålesund is a research town managed by Kings Bay, which is responsible for the Ny-Ålesund planning area. We also have an Italian Station Leader who monitors the situation and records each human activity near the sampling areas in the logbook. For the 2019 sampling period, no ordinary or extraordinary maintenance to fittings or welds' pipelines were recorded, neither paintings nor particular event that may justify a Pb release. Furthermore, the sampling area is a clean area, where no snowmobiles can have access.
- R-30 l.373-375: trace elements (and ions) concentration time series would be best plotted with wind direction and speed.
  - A-30 We thank the referee for highlighting this point. We added wind direction and speed against concentration time series, as suggested.
- R-31 l. 388: "dichlorination [...] at the snow-atmosphere interface [...]", please describe the process invoked here.
  - R-31 The de-chlorination is a reaction that occurs between sea-salt particles, containing NaCl, and HNO3, H2SO4, or organic acids to release gaseous HCl (Zhuang et al., 1999 and reference therein). In particular, NOx transforms into gaseous nitrous and nitric acids, which later react with NaCl in sea-salt aerosols to form NaNO3 and HCl in the so-called chloride depletion reaction (Savoie and Prospero,

1982; Harrison and Pio, 1983; Wall et al., 1988; Wu and Okada, 1994; Kerminen et al., 1997; Harrison et al., 1994; Pakkanen, 1996). SO2 oxidation and to a lesser extent H2SO4 vapor condensation on seasalt aerosols can also lead to chloride depletion (Sievering et al., 1991, 1995). We added a brief explanation within the main text (L 574-576).

- R-32 I. 389: "negative correlation", please provide a coefficient and p value. Thanks for the suggestion. We provided the values for the correlation (actually, very low, not negative) and p value (L 578).
- R-33 l.392: "appear positively correlated", where? Please prove a coefficient and p value. Again, we directly specified them at L 580.
- R-34 I.399-410: are the calculations based on the full campaign data or just on 1 season of the campaign?

A-34 The calculations were based on the full dataset from all three sampling campaigns (2018-19, 2019-20, and 2020-21). This comprehensive approach allows us to account for seasonal variability and provides a more robust understanding of the sources and contributions of sulfate (SO42-) and other ionic species in the surface snow. By incorporating data from all seasons, we aimed to capture the full range of environmental conditions and potential influences affecting snow chemistry over the entire study period.

- R-35 l.410: "(2020-21)", do you mean "2019-20"?
   A-35 Exactly, this was an oversight, thanks for noting it. We changed to 2019-20.
- R-36 I.416-422: good explanation about MSA. Please discuss the wind direction at that time. A-36 We thank the referee for raising this important point. During the period of the MSA peak (April 2020), the wind directions seem to predominantly fall between 180° and 240° (south to southwest), as indicated by the orange and red dots. This suggests that the southerly and southwesterly winds could be contributing to the transport of marine aerosols, including those containing DMS (dimethyl sulfide), which is the precursor to MSA, from open ocean areas to the Arctic region (Ny-Ålesund). Additionally, sporadically but high-speed (> 5 m sec-1) north-northwest winds (0° to 60°) also occurred during that period. We added a short discussion about the wind directions within the manuscript (L 681-683).
- R-37 I.426: "a species known for high DMS production in 2019", which one? Please be specific here.
  A-37 The species recognized for its significant dimethylsulfide (DMS) production is *Phaeocystis pouchetii*, a phytoplankton known for its capacity to generate DMS in substantial amounts. We appreciate the referee for highlighting this point, and we have now included the species name in the manuscript.
- R-38 I.432: [ex-SO42-], why not using the biogenic SO4 fraction calculated earlier?
   A-38 We appreciate your insightful comments on our methodology for determining the anthropogenic contribution to non-sea-salt sulfate. Our approach utilizes the formula: [ex-SO42-] = [SO42-] (0.12 [Na+]) (0.175 [Ca2+]).

This method allows for a clear separation of sulfate contributions from sea salt (sodium) and mineral sources (calcium), enabling us to isolate the remaining sulfate as an indicator of anthropogenic inputs. Given that our analysis reveals biogenic sulfate contributions ranging from 0.15% (2018-19, 2020-21) to 0.38% (2019-20), we initially chose to focus on anthropogenic sources to better understand human impacts on the sulfate budget.

We also calculated the biogenic contribution separately using the methanesulfonic acid (MSA) loads, recognizing the significance of this pathway in sulfate formation. This dual approach allows for a comprehensive assessment without mixing the two sources.

While our current methodology aligns with standard practices in atmospheric chemistry, we acknowledge the value of your suggestion regarding the incorporation of the biogenic sulfate fraction in our calculations. As biogenic contributions are relatively minor, we opted for a focused analysis on anthropogenic sulfate; however, we are open to further discussions on this topic and appreciate your feedback to enhance the clarity and rigor of our study.

- R-39 I.433: "the already clarified reasons", which ones? It is not clear which reasons are referred to. A-39 We apologize for the lack of clarity regarding the "already clarified reasons." In the manuscript, we referred to the fact that Ca2+ concentrations were not measured in the samples collected during the second campaign due to instrumental limitations. This point was addressed earlier in the paper (sect. 2.2). We appreciate your attention to this detail and will ensure that our discussion more explicitly connects to this reasoning for improved clarity.
- R-40 I.434: Please give a reference for the coefficients employed in this equation.
   R-40 The ex-SO4 formula was introduced by Schwikowski et al. (1999). The coefficient 0.12 is based on the molar ratio of SO4 to Na+, while the value 0.175 represents the observed slope corresponding to the pre-industrial nssSO4/nssCa ratio in mineral dust found in snow.
- R-41 I. 436: "previous results show [...]", please report which were those results.
  A-41 In our manuscript, when we mentioned that "The obtained results showed a 50 up to 60% of anthropogenic contribution for the nss-SO42- input," we were referencing the findings of Amore et al. (2022). They reported that anthropogenic sulfate (Anthro-SO42-) was the most abundant component across all years in the study area, accounting for at least 50% all over the year, and covering more than 80% of the total sulfate budget during peak months, particularly from February to May (82.8% in GVB and 82.9% in ZEP). We will clarify this point in the text by including the specific results from Amore et al. (2022) to provide the necessary context for our findings.
- R-42 I.431-441: which season is considered here?
   R-42 We calculated the mean percentage contribution for each sampling season, thereby providing a comprehensive overview across the entire campaign year.
- R-43 l.440-441: "[...] suggesting a separate origin." Could it be a crustal origin?
   A-43 The nss-SO4, referred to here as having "a separate origin," is unlikely to originate solely from crustal origin. This is supported by the fact that nss-SO4 shows a positive correlation only with Mg (0.44, p < 0.05), which has an enrichment factor (EF) greater than 10, indicating a significant contribution from mixed sources.</li>
- R-44 l.445: "[...] rather than anthropogenic activities". What makes the authors rule out the anthropogenic source? Please explain specifically.
   A-44 Here, our intention was not to rule out the anthropogenic source, but only to highlight that the major link was observed with sea-salt ions and biogenic emissions. Maybe we can better clarify it in the text.
- R-45 l.461. Please explain the process involved in the production of Br with the formation of sea ice.
   A-45 The link between sea ice and Br is well studied and several references can explain the process (see for example Scoto et al. 2022, <a href="https://doi.org/10.1073/pnas.2203468119">https://doi.org/10.1073/pnas.2203468119</a>; Spolaor et al. 2013,

https://doi.org/10.5194/acp-13-6623-2013; Spolaor et al. 2014, <a href="https://doi.org/10.5194/acp-14-9613-2014">https://doi.org/10.5194/acp-14-9613-2014</a>, Vallelonga et al. 2021, <a href="https://doi.org/10.1016/j.quascirev.2021.107133">https://doi.org/10.1016/j.quascirev.2021.107133</a>). Briefly, the first-year sea ice surface is extremely reactive and, during spring, the Br can be emitted as a gas from the sea ice surface through the bromine explosion reaction (a reaction that involves the Br ions in the sea ice surface and the ozone and enriches the average concentration of this halogen in the atmosphere) causing also an enrichment of Bromine in the deposition and hence in the snow.

- R-46 Paragraph 4.4.: which season, which part of the dataset is used in the cluster analysis? We used the entire dataset for the HCA, encompassing all three sampling campaigns (2018-2021), in the cluster analysis to capture the full variability in sulfate sources and atmospheric processes across different seasons and years. By including all data points, we avoid potential bias that could arise from focusing on a single season or time period, and we ensure that the resulting clusters reflect the overall, long-term trends in sulphate contributions.
- R-47 I.468-475: The 2019 Pb peak and the 2020 Pb peak are two different events and probably not marking the same source input. The 2019 Pb could be of anthropogenic origin, while the 2020 Pb could be crustal. Therefore, it is important to try calculating the enrichment factor of the trace elements. The larger sea ice extent in Spring 2020 and a dust input (nutrient) could explain the peak in MSA (and the correlation between crustal trace elements and MSA).
  R-47 We thank the referee for highlighting this important point. In response, we calculated the crustal enrichment factors (EFs) of the trace elements, which provided valuable insights. In spring 2020, the EF for Pb was below 100, indicating a mixed origin with possible crustal contributions. However, the EF for Ni during the same period exceeded 100, pointing to an anthropogenic source. Therefore, while the increased sea ice extent and a nutrient (dust) input in spring 2020 could explain the MSA peak,
- R-48 I.518-520: this sentence is unclear. Please develop or explain.
   A-48 By adding the nss-SO4 and bio-SO4 to the HCA, we obtained slightly different results, with MSA grouped within the marine cluster. Therefore, we entirely revised this part of the discussions.

**Cited references:**

this may not be the sole factor at play.

Rhodes, R. H., Yang, X., & Wolff, E. W.(2018). Sea ice versus storms: Whatcontrols sea salt in Arctic ice cores?Geophysical Research Letters, 45,5572–5580. https://doi.org/10.1029/2018GL077403

Ruppel, M. M., M. Khedr, X. Liu, E. Beaudon, Sönke Szidat, Peter Tunved, Johan Ström et al. "Organic Compounds, Radiocarbon, Trace Elements and Atmospheric Transport Illuminating Sources of Elemental Carbon in a 300-Year Svalbard Ice Core." *Journal of Geophysical Research: Atmospheres* 128, no. 16 (2023): e2022JD038378.

Rankin, A. M., Wolff, E. W., & Martin, S. (2002). Frost flowers: Implications for tropospheric chemistry and ice core interpretation. Journal of Geophysical Research, 107(D23), 4683. https://doi.org/10.1029/2002JD002492

Beaudon, E. and Moore, J., 2009. Frost flower chemical signature in winter snow on Vestfonna ice cap, Nordaustlandet, Svalbard. *The Cryosphere*, *3*(2), pp.147-154.

Roscoe, H.K., Brooks, B., Jackson, A.V., Smith, M.H., Walker, S.J., Obbard, R.W. and Wolff, E.W., 2011. Frost flowers in the laboratory: Growth, characteristics, aerosol, and the underlying sea ice. *Journal of Geophysical Research: Atmospheres*, 116(D12).

Review of Impact of Arctic Amplification variability on the chemical composition of the snowpack in Svalbard (Spagnesi et al., 2024).

In this study, Spagnesi and coauthors present results of the chemistry of the snow in Ny-Alesund, Svalbard. The authors have generated detailed information of the chemical composition of the surface snow layer during a 3-year period from 2019-2022. The studied species include major ions, but also a number of further chemical elements. I'm not aware of a similar detailed data set concerning the chemistry of Arctic snow covering such a long period. In my opinion, however, the authors do not use this valuable information to advance our understanding concerning processes at the atmosphere-snow interface. Instead, they put their observations in the context of Arctic Amplification, which is my opinion impossible to justify with the available data. I describe my major concerns in the more major comments below. As a result, I do not recommend publication of the manuscript in its current form.

**Major comments**

- R1- My understanding of the Arctic Amplification (AA) is seriously different as it is presented by the authors and how it is used throughout the manuscript and even in the title. As mentioned by the authors themselves, "AA is recognized as an inherent characteristic of the changing global climate system, with multiple intertwined causes operating on a spectrum of spatial and temporal scales." (L. 55f). I would even remove the word "changing" since I think AA is a part of the current global climate system, but becomes only visible if the climate system is changing. However, the authors seem to suggest that AA is turning on and off (e.g. "Arctic Amplification (AA) period", L. 40; "similar to the pre-AA conditions", L. 41; "appearing more similar to Arctic type than Arctic Amplification conditions", L. 277) suggesting that years with higher observed temperatures are due to AA, while colder years are not. However, I think it is obvious that such differences during a 3-year period are due to interannual variabilities of the climate system. I do not exclude that these meteorological differences and, thus, differences in the chemical composition of the snow are enhanced by the AA, but the data presented in this study do not contribute to this issue since time series of 3 years are not sufficient. In my opinion, up to chapter 4.1 the manuscript should be revised to put the relationship between the observations and the AA in the right context.
- A1- We greatly appreciate the referee's feedback. We realize that some of our wording may have unintentionally suggested that AA is only present during warmer years. To clarify, our intention was not to imply that AA operates by turning on and off, but rather to discuss how the variability we observed could be influenced by broader AA-driven processes. We fully agree that the observed differences in temperature and snow chemistry over a short period (e.g., 3 years) can likely be influenced by interannual variability. However, we also believe that AA is a significant background factor influencing the Arctic climate system, even on shorter timescales. In light of this, we have revised the manuscript to better contextualize our observations within the broader framework of Arctic Amplification.
- R2- The authors conclude chapter 4.1 with the following statement: "These findings highlight the complex interplay between atmospheric patterns (AO and wind speed), local climate (temperature and sea ice extent), and oceanic conditions (SST, salinity), showing similar ionic and elemental trends in surface snow for wind, sea ice, and SST counterbalanced conditions." (L. 352ff). I think that such a statement is superfluous here. Since several decades, it is known that the chemical composition of the snow is impacted by a complex interplay of the atmospheric composition and a number of depositional and post-depositional processes. These factors apparently depend on atmospheric and

marine processes, on local and regional conditions and so on. This should be the starting point of such a study and not the result or the conclusion. It would actually be important to analyze what new information the new observations can deliver (or not). A-2 We see the point of the referee, and agree that the complex interplay of atmospheric and marine processes on snow chemistry has been established in previous research. Our intention with this statement was not to present a novel conclusion but rather to summarize and contextualize the observations we made during our study, particularly in relation to the specific combination of conditions observed during the 2020 season (i.e., strong positive AO, wind speed anomalies, and variations in sea ice extent and SST). However, since we agree that this sentence may sound redundant, we have better revised this part, focusing on new insights offered by our observations.

- R3- Therefore, the interesting part of the manuscript starts in my opinion only with chapter 4.2. Here the authors start exploiting their unique chemical data to analyze different processes by looking at concentration ratios, enrichments, depletions and so on. Unfortunately, this is presented in very general terms mostly with annual ratios or average ratios for the entire 3-year period and without a graphic representation of any of this data. For example, a plot of the MSA vs. nss-SO42- with all samples should make the differences, briefly discussed in chapter 4.2, much more visible. Taking such plots as a starting point would in my opinion demonstrate much better the variability between (and within?) the 3 different years of the observations. Such an analysis could then be used for a study of the meteorological processes and conditions that contributed to the observed differences. For example, the authors state that to "clarify the origins of Pb peaks recorded between winter and spring 2019 further investigations are needed, which goes beyond the scope of this study." (L. 373ff). I suggest that such issues could actually rather be the scope of the manuscript. The Anonymous Referee #1 gives some further examples what could be studied by looking at specific ratios.
  - A-3 We appreciate the referee's comment and fully understand the point raised. We initially considered that Fig. 2 would effectively represent the trends of all ions and trace elements across the three sampling campaigns. However, based on the valuable suggestions from both referees, we recognize the importance of comparing these trends with meteorological data (e.g., precipitation, wind speed, wind direction). In response, we have incorporated this meteorological information into Fig. 2 to enhance our analysis. Additionally, to highlight the significance of the enrichment factor (EF) calculations and their role in identifying elemental sources, we have included an EF graph (Fig. 3) in section 4.1, which enriches the previous discussion on Pb.
- R-4 I am very concerned about the snow sampling strategy and the justification of this strategy by
  the authors. They state that the "surface snow was sampled within the upper 3 cm, as this is the snow
  layer most influenced by the aerosol-cryosphere exchanges, and, in case of snowfall, by deposition
  (Spolaor et al., 2018, 2021b). This choice also minimised the effect of different physical snow
  conditions (density, crystal shape and size)." (L. 102ff). While it is true that the surface layer is mostly
  impacted by the interaction with the atmosphere, I do no understand, how this strategy minimizes
  the impact of processes within the snow or in the snowpack on the observed concentrations.
  - A-4 There are several scientific and logistical reasons behind the sampling strategy we adopted. The decision to sample the upper 3 cm of the snowpack is primarily based on our interest in capturing what is being deposited from the atmosphere through dry and wet deposition. The upper 3 cm is the most representative of these atmospheric inputs.

In the case of a dry period (without new snowfall), dry deposition will affect only the surface layer (the "snow skin layer") rather than the interior of the snowpack. Sampling a deeper layer (e.g., 10–15 cm) could smooth out the impact of surface dry deposition, diluting its signal. For instance, if the density and concentration of element X are uniform across the upper 3 cm and 15 cm layers, the total mass of element X in the upper 15 cm would be much higher, thereby minimizing the

detectable contribution of dry deposition. By focusing on the upper 3 cm, we reduce the overall mass of element X, making the effect of dry deposition more significant and easier to detect.

During a snowfall, we assume that the elemental composition of the snow remains consistent throughout the event, based on the premise that the atmospheric load remains steady (as proposed in Schupbach et al. 2021). Although we recognize that this assumption may not always hold, there could be variations in the elemental load at the beginning versus the end of a snowfall event, this approach still works well, particularly during weak snowfall events (<5–10 cm of accumulation). For more intense snowfall events (>5–10 cm of accumulation), there may be a risk of underestimating or overestimating elemental concentrations, but overall, we consider the fresh upper layer to be representative of the event.

Another consideration is the practicalities of a long sampling campaign (spanning the entire winter season). The strategy we use must be robust and simple for the operator at Dirigibile Italia Station, minimizing the chances of error due to improper sampling. Ideally, sampling the entire snowpack daily would provide the most comprehensive data, but this would require significant effort and manpower. Furthermore, such frequent sampling could compromise the reliability of the area for future research due to excessive disturbance.

In the event of a winter melting episode, the integrity of the full snowpack may be compromised, but while the internal stratigraphy is lost, we are still able to capture changes in surface deposition after rain or melting events. Although studying the full snowpack offers insights into how the snowpack evolves in response to deposition and melting episodes, our focus is mainly on evaluating the seasonal differences in elemental dry and wet deposition

- R-5 This becomes obvious if for example between two samplings the surface snow layer is removed due to erosion during blowing snow conditions. In that case, a deeper, older snow layer can be exposed at the surface and will be collected with the latter sample. This can have a serious impact on the presented time series. It is well known that such events happen regularly at Ny-Alesund and in fact the meteorological data indicate that such events (e.g. reduction of the snow height at below-zero temperatures and during periods with high wind speeds) occurred during the 3-year period.
  A-5 This is certainly possible, but other factors need to be considered, and the exposure of older snow layers is not as straightforward as it might seem. Wind erosion can remove the upper snow layers and expose older layers, but this process also occurs downwind from the sampling site. This means that the wind can transport and deposit snow from adjacent areas back onto the sampling site. Additionally, wind has the effect of compacting the upper snow layer, which makes wind erosion more efficient at the beginning of a snow event than towards the end. It's important to note that we are working with a natural snowpack, which means dealing with the inherent complexities and variability of real environmental conditions. The matrix we are sampling from is
- R-6 I suggest to study in detail such events to look at the impact on the time series. In this context I also recommend to carefully revise statements like "winter seasons showed the higher average loads" (L. 254), which gives the impression that between winter and spring the total load of the snowpack is reduced, which is unlikely. The statement above concerns only the load of the snow surface layer (or more precisely: the top 3 cm of the snow). Unless the impurities are washed out or removed by erosion due to blowing snow, it is likely that the total load of the snowpack increases from winter to spring with the increase of the SWE.

subject to various factors, such as wind compaction, transport, and deposition, which complicate

a simple interpretation of wind erosion or deposition effects.

A-6 We agree with the reviewer's comment, and to clarify, we are indeed referring to the higher deposition rate of a specific element (e.g., sodium). We are not referring to the total load of the entire snowpack. If we were considering the whole snowpack, the total load would likely increase

with the snow water equivalent (SWE). Instead, we are focusing on the deposited load within the resolution we adopted, which shows a higher deposition rate in a specific season. This is consistent with observations made in other Arctic locations, as well as in Svalbard (e.g., sodium) (e.g., Macdonald et al., 2018; Jacobi et al., 2019).

**Minor comments**

- R-7 L. 46: I don't understand the meaning of a "wintry stratospheric polar vortex".
   A-7 We referred to "wintry stratospheric polar vortex" to stress the seasonal aspect, but we are aware that these large-scale cyclonic circulation forms over the polar region during winter, making the "wintry" distinction potentially unnecessary.
- R-8 L. 83ff: "the chemistry of Arctic snow and the exchange of inorganic species between cryosphere and atmosphere have been the subject of a relatively small number of studies or of specific events (Dommergue et al., 2010; Spolaor et al., 2013, 2019; Barbante et al., 2017)." I disagree with this statement. The exchange between the atmosphere and the snow has been studied for decades in the Arctic. There have been studies at least at Barrow, Summit Greenland, Alert, Sodankylä, or over the Arctic Ocean during the MOSAiC expedition. A detailed literature research would probably deliver studies from further sites. Even at Ny-Alesund and in the area around there have been many more studies than these few cited, e.g. Beine et al., ACP 3, 335-346, DOI 10.5194/acp-3-335-2003, 2003; Björkman et al, Tellus B 65, DOI 10.3402/tellusb.v65i0.19071, 2013; Jacobi et al., ACP 19, 10361-10377, DOI 10.5194/acp-19-10361-2019, 2019 to name just 3 examples from the last 20 years.
  - A-8 Our intention here was to underscore the limited scope of research specifically addressing the temporal and spatial variability of these exchanges at Ny-Ålesund and its immediate surroundings. However, we agree with the referee, and we revised the text to more accurately reflect this view and include references to additional relevant studies, such as those cited by the referee, to strengthen the discussion.
- R-9 L. 152f: "However, Ca2+ was not measured within the samples collected during the second campaign due to instrumental limitations". How was this treated since later on data for calcium are presented for all three winter seasons. Ca was also determined with the ICP-MS. Is this data used instead for calcium or only during the second winter?
  - A-9 Due to instrumental issues encountered during the analysis of the second batch of samples (2019-20), we opted to focus our discussion on the total Ca concentrations measured with ICP-MS across all three sampling campaigns.
- R-10 L. 217: "Comparison between concentration trends at Gruvebadet and Ny-Ålesund". Don't the authors compare concentrations? Moreover, my understanding is that the Ny-Ålesund data are not used for the further analysis. If so, do they need to be in the manuscript? I understand the interest to publish the data, but wouldn't it be sufficient to put them in the supplement?
  - A-10 The comparison between Gruvebadet and Ny-Ålesund concentrations is crucial for assessing the spatial variability of the measured parameters, which is a key aspect of our study. While the Ny-Ålesund data are not used directly in further analyses, they provide important context and serve as a reference point to highlight local-scale differences in concentrations. For this reason, we believe it is important to include the comparison in the main manuscript, as it strengthens the interpretation of our findings.
- R-11 L. 223f: "with few isolated peaks for sea salt and crustal tracers present in the Ny-Ålesund record from November to February, following positive temperature anomalies and precipitation events (Fig. S2)." In Fig. S2 I identify longer periods with elevated concentrations in Na+ and Ca2+ at Ny-Alesund

in November, January, and February. However, in January there was no precipitation with low temperatures; for Ca2+ there is a further very strong peak in early February, but no precipitation and low temperatures. It seems that such a general statement is not supported by the data. Moreover, why would high precipitation lead to high concentrations in the snow?

A-11 We agree that the statement regarding the connection between sea salt and crustal tracer peaks with positive temperature anomalies and precipitation requires further clarification. While we observed peaks in Na+ and Ca2+ concentrations at Ny-Ålesund during November and February, some occurred under low temperatures and without significant precipitation (e.g., January and early February). This suggests that additional processes, such as long-range transport, wind-driven deposition, or post-depositional effects like sublimation of the water matrix (Ginot et al., 2001), could have contributed to these peaks, even in the absence of immediate precipitation. We revised the text to reflect these potential factors and clarify that high precipitation does not always directly lead to high concentrations, as other atmospheric processes may be involved.

- R-12 L. 233f: "referring to ionic loads (mg m-2) instead of concentrations (ng g-1), to highlight the seasonal trends of specific tracers. The ionic load is calculated as ionic concentrations multiplied by the density and the depth of sampled strata." See also major comment above, unclear why the loads of the top 3 cm layer are better placed to show seasonal trends, since I assume that "the depth of sampled strata" is actually always 3 cm. Moreover, the manuscript later on refers still at some occasions to concentrations (e.g. "A singular case is represented by Pb, with remarkable trend concentration...", L. 359; caption of figure S3).
  - A-12 Thanks for noting this incongruence. We have modified the manuscript to maintain consistency by using flux throughout. Although the sampled strata remain uniform, snow density can vary due to factors such as wind drift, snow aging, melting, or fresh snowfall. Since we consistently sample the same depth, the only parameter, apart from concentration, that can change is the density. For example, if a sample shows 5 ng/g of element X with a density of 0.25 kg/L, it differs in total mass concentration from a sample with 5 ng/g but a density of 0.35 kg/L. Although the concentrations are the same, the mass of element X in the upper 3 cm layer would be lower in the first case than in the second. This is why we calculate and use the flux instead of concentration, as it allows us to evaluate the total mass of element X in the upper snow layer more accurately.
- R-13 L. 255: "Higher values of sea salts species were found in autumn 2020, but less snow accumulation was recorded during that period (Fig. 1)." According to Table S3 the precipitation average for autumn 2020 was 4.91 mm (Is this actually in mm/day?), which is the second highest value in this Table. This does not seem to support this statement.
  - A-13 We agree that the precipitation average for autumn 2020, listed as 4.91 mm in Table S3, does not directly support the original phrasing regarding "less snow accumulation" during this period. To clarify, we have highlighted that despite the relatively high precipitation amount (seasonal mean, in mm), the lower snow accumulation in autumn 2020 (17.25 cm), compared to other seasons with similar or lower precipitation levels (e.g., spring 2020), could be attributed to a combination of factors such as higher wind speeds and more frequent days with moderate or strong winds during that period, potentially leading to snow redistribution or sublimation.
- R-14 L. 286: "promoting the formation of sea ice in Kongsfjorden,": the temperatures shown in Tab. S4 are not sufficiently low for the formation of sea ice.
  - A-14 We acknowledge that the temperatures at 10 meters depth shown in Table S4 are not low enough by themselves to promote sea ice formation. However, the formation and persistence of

sea ice in Kongsfjorden during 2020 were likely influenced by a combination of factors. While the seawater temperature at 10 meters was above freezing, strong stratification in the fjord, driven by freshwater input from glaciers, likely created a fresher, colder surface layer where sea ice could form. Additionally, cold atmospheric conditions during the 2020 winter and spring seasons would have contributed to surface cooling, facilitating ice formation and prolonging its presence through the season. We clarified the role of these factors in the manuscript.

- R-15 L. 289ff: "Salinity data also revealed modest fluctuations across the considered seasons, showing a decrease of 0.35 psu in 2020 relative to 2019, and a decrease of 0.1 psu compared to 2021." Why are these observations relevant? Only for the sea ice or also for the aerosol formation?
  A-15 We mostly agree with this observation, since the reported salinity fluctuations are modest and their significance becomes apparent only if considered alongside colder temperatures and wind conditions, as they play a key role in the dynamics of sea ice formation during that period. In general, fluctuations in salinity are relevant not only for sea ice formation but also for aerosol formation. Lower salinity, which can be driven either by freshwater inputs (e.g., glacial melt), as well as precipitation, can influence the sea ice formation process, which in turn affects the release of sea-salt aerosols and biogenic emissions from the sea ice surface. However, in this case, they do not add great advance in the interpretation of the sea ice formation.
- R-16 L. 293ff: "campaigns conducted from 2018 to 2021 confirmed the dominance of sea salt input in the surface snow of Svalbard, likely due to the proximity of the Kongsfjord (Barbaro et al., 2021). The dominant ions are Na+, Cl-, and SO42-, likely associated with the scavenging precipitation of marine aerosol": I think that this can be considered as facts, so that the word "likely" is not needed here.

A-16 We agreed with the referee's comment and we deleted "likely" from these sentences.

- R-17 L. 314ff: "was recorded in January-March 2020 (Lawrence et al., 2020; Dethloff et al., 315 2022), featuring as an outlier in the historical timeseries 1950-2023 reported by the NOAA service." In Fig. S6 only Jan and Feb are indicated as outliers.
  - A-17 The AO value in March 2020 was still above the mean but did not exceed the threshold for statistical outliers in the long-term historical data. Despite this, the persistence of a positive AO phase from January to March 2020 remains significant when considering the context of the entire season. We better clarified this aspect within the main text.
- R-18 L. 376ff: "Other backward trajectories (Fig. S8) for Ny-Ålesund area (78.92° N, 11.89° E) appear mostly in line with literature findings (Platt et al., 2022; Vecchiato et al., 2024), showing three main seasonal characters: a prevalent mass movement from ice-covered Central Arctic Ocean, Kara Sea, and Greenland Sea during autumn, a main provenance from Central Arctic Ocean and Kara Sea during winter, and a predominant trajectory from Northern Canada in addition to air masses arriving from Arctic Ocean and Kara seas during spring." If this confirms previous results, why is it important to describe here?

A-18 We see the point of the referee. However, we retain that detailing seasonal trajectory patterns enhances the significance of our findings by linking them to broader climate discussions.

R-19 Fig. S6: A complete reference for the presented data is missing.

A-19 We have added a proper reference link for the presented data, as the initial attribution was made solely to NOAA without a specific citation.

---

## Referee Report (RR1)

This study presents measurements of trace elements and major ions in surface snow collected during three field campaigns from 2018 to 2021. The authors report higher concentrations of marine-origin species in late spring 2020, likely driven by specific meteorological and oceanic conditions. The results also show a strong correlation of impurities in Ny-Ålesund during cold seasons. The manuscript lacks a clear explanation of how these findings contribute to our understanding of climate change. The introduction sets up climate relevance, but the discussion does not adequately follow through. A more focused and contextualized discussion is necessary to justify the broader implications claimed by the authors. The paper can be published after addressing these comments.

**Major comments.**

R-1. Line 89: You mention that this study contributes to understanding trace element and ion interactions in the context of recent climatic changes. However, this connection is not clearly addressed in Section 5, *Summary and Conclusion*. If the stated goal is "to enhance the understanding of these interactions, particularly in the context of recent climatic changes," then the conclusion should explicitly discuss how your findings support or inform that objective. As it stands, the broader relevance to climate change is implied but not directly explained. This should be clearly articulated before the manuscript can be considered for publication.

R-2. Line 114: Did you collect and analyze background concentrations or include any blank/control samples during sample handling? It is important to clarify how you ensured that the snow samples were not contaminated during collection, transport, or analysis. Please explain the procedures used to confirm sample integrity and rule out possible contamination.

R-3. Line 150: It would significantly improve the clarity of the manuscript to include a map showing the sampling locations. Since the sampling was conducted across multiple sites during three separate field campaigns, a visual representation in the main text (not only in the supplementary materials) would help readers better understand the spatial context of the study.

R-4. Line 202: The manuscript should clearly explain the rationale for selecting a 6-hour back-trajectory interval with a propagation time of 120 hours.

R-5. Line 238-239: You state that "the difference in concentration trends appears very low or negligible, except for sporadic peaks in sea salt and crustal tracers present in the Ny-Ålesund record from November to February." However, the term *negligible* needs to be supported with quantitative data. Please specify the concentration values and the average

concentration differences compared to the November–February period to substantiate this claim.

R-6. Line 405: Please provide appropriate references to support the statement that enrichment factors (EF) below 10 indicate a crustal origin of the elements.

R-7. Line 520: You refer to correlation results multiple times throughout the manuscript, but you do not indicate which figures or tables support these findings. Please clearly reference the relevant figures or tables in the text. If these results are not currently included, they should be added to the supplementary information to support the discussion.

R-8. Line 554: As previously mentioned, you need to provide specific values rather than stating that "the errors associated with the EFs are quite high."

**Minor comments.**

R-9. Line 210: Provide the full name for the acronym NCEP/NCAR at its first mention in the text to ensure clarity for all readers.

R-10. Line 233: There is a missing period at the end of number 3. Please add it to maintain proper punctuation.

R-11. Line 271: In Table 1, "Total" should be capitalized to maintain consistency with proper noun formatting. Change "total" to "Total."

R-12. Line 396: Instead of using vague terms like "slightly above," I recommend providing the exact number to improve clarity and precision in your results.

R-13. Figure S2: In the figure, the plots for Pb and Ca2+ slightly overlap, particularly where their highest concentrations coincide. To improve clarity, consider adjusting the plot style to avoid confusion in interpreting the peaks.

R-14. Figure S3: You need to explain why the graph includes a gray background for the periods from autumn 2018 to autumn 2020 and from spring 2019 to spring 2021.

R-15. Table S2: The "k" in "Km2" should be lowercase. Correct it to "km2" to follow proper SI unit formatting.

R-16. Figure S7: The figure needs to be provided in higher resolution. The text and numbers under the Ice Categories are difficult to read in the current version.

---

## Author Response (AR2)

**Reviewer #1**

This study presents measurements of trace elements and major ions in surface snow collected during three field campaigns from 2018 to 2021. The authors report higher concentrations of marine-origin species in late spring 2020, likely driven by specific meteorological and oceanic conditions. The results also show a strong correlation of impurities in Ny-Ålesund during cold seasons. The manuscript lacks a clear explanation of how these findings contribute to our understanding of climate change. The introduction sets up climate relevance, but the discussion does not adequately follow through. A more focused and contextualized discussion is necessary to justify the broader implications claimed by the authors. The paper can be published after addressing these comments.

A: We thank the reviewer for their helpful comments. As suggested, we have improved the manuscript by clarifying the discussion of our findings in the context of climate change. Specifically, we now better highlight how the observed seasonal variability and impurity correlations relate to Arctic atmospheric and oceanic changes. We believe these revisions strengthen the manuscript and address the reviewer's concerns.

**Major comments.**

**R-1. Line 89:** You mention that this study contributes to understanding trace element and ion interactions in the context of recent climatic changes. However, this connection is not clearly addressed in Section 5, Summary and Conclusion. If the stated goal is "to enhance the understanding of these interactions, particularly in the context of recent climatic changes," then the conclusion should explicitly discuss how your findings support or inform that objective. As it stands, the broader relevance to climate change is implied but not directly explained. This should be clearly articulated before the manuscript can be considered for publication.

**A-1.** We see the point raised by the Reviewer 1, and we agree that the relevance to climate change has to be better explained in Summary and Conclusion. For this reason, we modified the text from L620, changing this statement: "In fact, sea ice has a role in concentrating, storing, and releasing marine species, as well as influencing atmospheric and oceanic processes that affect their production and distribution." with "These results provide direct evidence of how sea ice extent modulates the storage, release, and transport of marine-derived impurities, thereby influencing snow-atmosphere chemical exchange processes under varying climatic conditions." Furthermore, from L627, the following paragraphs:

"Therefore, our results highlighted a close dependence of high concentrations of impurities found in the snowpack at Ny-Ålesund on meteorological conditions, especially during cold years, when the production of sea spray related aerosol likely derives by a larger extension of sea ice and stronger local Arctic circulation. The identification of geogenic, marine, and anthropogenic sources in the snowpack was allowed by a chemometric approach (HCA), which clarified the EFs results. The back trajectories analysis revealed distinct seasonal air mass patterns. During fall and winter, air mass predominantly originated from Northern Canada in addition to air masses arriving from Arctic Ocean and Kara seas during spring. On the contrary, no prevalent mid-latitude air currents were revealed in spring as expected, considering the period of the three sampling campaigns (2018-2021). These findings offer new insights into how specific meteorological and oceanic conditions, such as sea ice extent, wind speeds, and Arctic Oscillation phases, influence the chemical composition of the snowpack in Svalbard, particularly within the context of Arctic Amplification."

have been changed with: "Such findings illustrate how large-scale atmospheric circulation anomalies associated with Arctic Amplification can significantly alter the deposition patterns of both natural and anthropogenic species in the snowpack. This is particularly evident especially in cold years, when the production of sea spray related aerosol likely derives by a larger extension of sea ice and stronger local Arctic circulation. The identification of geogenic, marine, and anthropogenic sources in the snowpack was allowed by a chemometric approach (HCA), which clarified the EFs results. The use of chemometric techniques (HCA) and back-trajectory analysis enabled a clearer attribution of sources and transport pathways, improving the interpretation of snow composition in relation to meteorological drivers. Specifically, the distinct seasonal air mass patterns revealed, characterised by dominant Arctic-origin air masses in fall and winter and a lack of expected mid-latitude inputs in spring, underscore the changing dynamics of snow-atmosphere interactions in a warming Arctic. Overall, these insights advance our understanding of how recent climatic anomalies, such as altered sea ice extent, shifts in Arctic Oscillation phases, and stronger polar vortices, modulate the chemical composition of the snowpack in Svalbard. Our findings highlight the sensitivity of snow-atmosphere exchanges to both local and large-scale climatic processes, offering important context for interpreting snow chemistry trends in a rapidly changing Arctic environment."

We think that a clearer explanation has been provided to the readers in this way.

**R-2.** Line 114: Did you collect and analyze background concentrations or include any blank/control samples during sample handling? It is important to clarify how you ensured that the snow samples

were not contaminated during collection, transport, or analysis. Please explain the procedures used to confirm sample integrity and rule out possible contamination.

**A2.** We appreciate Reviewer1's comment and agree that the procedures to ensure sample integrity and minimise contamination should be clarified. To assess potential contamination during sampling and transport, we collected field blanks using metal-free vials (Avantor, VWR Centrifuge Tubes, CHN). Some of these vials were opened to ambient air at the sampling site for a few minutes without collecting any snow, then sealed and transported to the Ny-Ålesund laboratory, where they were filled with 2% HNO3.

In parallel, we prepared analytical blanks by following the same procedure (vials opened to air but without snow sampling), then sealed and transported to Venice, where they were filled with 2% HNO3 and ultrapure water (UPW) from the Venice lab. Both set of blanks were analysed to detect any background contamination and were consistently below LODs or one order of magnitude lower than the lowest concentration detected for all analytes.

We have integrated this clarification into the main text to better explain the control measures used to ensure sample integrity throughout collection, transport, and analysis: "To assess potential contamination during sampling, handling, and transport, field blanks were collected during each campaign. Metal-free vials (Avantor, VWR Centrifuge Tubes, CHN) were opened to ambient air at the sampling sites for a few minutes without collecting snow, then sealed and transported to the Ny-Ålesund laboratory. There, they were filled with 2% HNO3 and stored under the same conditions as the snow samples. In parallel, analytical blanks were prepared by opening vials to air, sealing them, and transporting them directly to Venice, where they were filled with 2% HNO3 and ultrapure water from the laboratory. Both field and analytical blanks were analyzed alongside the snow samples, confirming that background contamination levels were below detection limits for all target analytes."

- **R-3. Line 150:** It would significantly improve the clarity of the manuscript to include a map showing the sampling locations. Since the sampling was conducted across multiple sites during three separate field campaigns, a visual representation in the main text (not only in the supplementary materials) would help readers better understand the spatial context of the study.
- **A-3.** We agree with the Reviewer's suggestion, and we included a map showing the sampling locations in the manuscript, moving the modified Fig. S1 from the Supplementary materials.
- **R-4.** Line 202: The manuscript should clearly explain the rationale for selecting a 6-hour backtrajectory interval with a propagation time of 120 hours.

- **A-4.** Authors welcome the suggestion of Reviewer1 and implemented this part in the manuscript. L201-210: "Back-trajectories were calculated every 6 h, with a propagation time of 120 h for each sampling period. The choice of a 6-hour interval for the calculation of back-trajectories allows for the capture of temporal variability in air mass origins over the day, which is particularly important in polar regions where atmospheric circulation patterns can change rapidly. This temporal resolution strikes a balance between computational efficiency and the need for sufficient detail to characterise the variability in source regions during each sampling period. The propagation time of 120 hours was selected to provide an adequate temporal window to trace long-range transport pathways that influence air mass composition at Ny-Âlesund. This configuration is consistent with previous studies on atmospheric circulation in the same site (Feltracco et al., 2021)."
- **R-5.** Line 238-239: You state that "the difference in concentration trends appears very low or negligible, except for sporadic peaks in sea salt and crustal tracers present in the Ny- Ålesund record from November to February." However, the term negligible needs to be supported with quantitative data. Please specify the concentration values and the average concentration differences compared to the November–February period to substantiate this claim.
- **A.5** We modify the sentence as suggested by Reviewer1: "the difference in concentration trends appears **not statistically significative (p values < 0.05, Wilcoxon test)** for all the analysed species, except for sporadic peaks in sea salt and crustal tracers present in the Ny-Ålesund record from November to February".
- **R-6. Line 405:** Please provide appropriate references to support the statement that enrichment factors (EF) below 10 indicate a crustal origin of the elements.
- A-6. We added the appropriate references here, as suggested: Wedepohl, 1995; Gabrieli et al., 2011
- **R-7. Line 520:** You refer to correlation results multiple times throughout the manuscript, but you do not indicate which figures or tables support these findings. Please clearly reference the relevant figures or tables in the text. If these results are not currently included, they should be added to the supplementary information to support the discussion.
- **A-7.** We thank the Reviewer for noting the lack of a supporting figure/table. We added Fig. 5 in the main manuscript.
- **R-8. Line 554:** As previously mentioned, you need to provide specific values rather than stating that "the errors associated with the EFs are quite high."

**A-8.** We see the point of the Reviewer. However, our intention here was not to provide specific values, but to highlight that the EF calculations may reflect larger uncertainties than the HCA method. For this reason, we decided to modify the sentence with the following statement: "This apparent discrepancy may reflect the relatively larger uncertainties typically associated with EF calculations, which can inherit errors from the choice of reference element, assumptions about crustal composition, and variability in background concentrations, compared to the more integrative approach of HCA (e.g., Reimann and de Caritat, 2000)."

**Minor comments.**

- **R-9. Line 210:** Provide the full name for the acronym NCEP/NCAR at its first mention in the text to ensure clarity for all readers.
- **A-9.** We thank the Reviewer1 for raising this point. We clarified the acronym in the text. L217-218: "National Centers for Environmental Prediction/National Center for Atmospheric Research (NCEP/NCAR)".
- **R-10.** Line 233: There is a missing period at the end of number 3. Please add it to maintain proper punctuation.
- **A-10.** We added the missing period to maintain proper punctuation, as kindly suggested.
- **R-11. Line 271:** In Table 1, "Total" should be capitalized to maintain consistency with proper noun formatting. Change "total" to "Total."
- **A-11.** We capitalized "Total" in Table 1, as suggested.
- **R-12.** Line 396: Instead of using vague terms like "slightly above," I recommend providing the exact number to improve clarity and precision in your results.
- **A-12.** We agree with the point raised by the Reviewer1, and we modified the vague expression with "(i.e., EF = 26)."
- **R-13. Figure S2:** In the figure, the plots for Pb and Ca2+ slightly overlap, particularly where their highest concentrations coincide. To improve clarity, consider adjusting the plot style to avoid confusion in interpreting the peaks.
- **A13.** We thank the Reviewer for raising up this point and we adjusted the plot style to avoid confusion in interpreting the peaks.

**R-14. Figure S3:** You need to explain why the graph includes a gray background for the periods from autumn 2018 to autumn 2020 and from spring 2019 to spring 2021.

**A-14.** We added the explanation in the caption: "The grey areas correspond to the distinct snow seasons."

**R-15. Table S2:** The "k" in "Km2" should be lowercase. Correct it to "km2" to follow proper SI unit formatting.

**A-15.** We put "Km2" in lowercase to follow proper SI unit formatting.

**R-16. Figure S7:** The figure needs to be provided in higher resolution. The text and numbers under the Ice Categories are difficult to read in the current version.

**A16.** We agree with the Reviewer1, and we provided a higher resolution figure (ex Fig. S7, now Fig. S8).

**Reviewer #2**

Note from the Authors: We would like to kindly inform that Reviewer 2 appears to have based their evaluation on an earlier version of the manuscript (the preprint uploaded before the first round of revisions). As a result, a few discrepancies between the lines and content of the text were identified during the revision process. Additionally, some comments referred to figures and text that had already been modified. Nevertheless, the Authors have made every effort to address all the comments thoroughly and would like to express their sincere thanks to the Reviewer for the valuable and insightful suggestions.

Review of the manuscript entitled "Impact of Arctic Amplification variability 1 on the chemical composition of the snowpack in Svalbard"

This is an interesting paper reporting a lot of data on snow composition from Svalbard. While many papers have already been published about this, the novelty of the present work is the attempt to link compositional data obtained over a 3 year period and meteorological conditions. This is definitely the added value of the manuscript. The idea is interesting and can potentially deepen our understanding of snow chemistry in the Arctic environment, but in my opinion it is not very well developed. **The weakest aspect of the work is the link between snow data and meteorology, which is now only qualitative.** The discussion is made reporting things like: the season of that year was coldest and so sea-ice related elements were higher. Since the authors have in their hand a bunch of detailed meteorological data, **I would really expect to find a correlation study between snow composition and meteorological variables.** Adding this kind of discussion would really improve the novelty of this work and increase its scientific significance.

- R1 In general I would try to be more quantitative across the entire manuscript, applying statistical tests to highlight the similarity of dissimilarity of data recorded in different seasons or years.
- **A1.** We really appreciate this valuable suggestion from Reviewer 2, and we try to be more quantitative across the entire manuscript. We applied statistical tests (Kruskal-Wallis, Dunn, Spearmann correlation) to highlight the similarity/dissimilarity of data (see answer to R3).
- R2 One other thing concerns the comparison between data gathered at GSRS and Ny-Alesund. In the end the authors decide to focus only on GSRS data. I am wondering whether deciding not to include Ny-Alesund data would be a good choice. In the end all the discussion about Ny-Alesund data is rather useless to the aim of the work.

- **A2.** The comparison between Gruvebadet and Ny-Ålesund concentrations is crucial for assessing the spatial variability of the measured parameters, which is a key aspect of our study. While the Ny-Ålesund data are not used directly in further analyses, they provide important context and serve as a reference point to highlight local-scale differences in concentrations. For this reason, we believe it is important to include the comparison in the main manuscript, as it strengthens the interpretation of our findings. However, since 2019 renovation activities at the Ny-Ålesund site have limited the reliability of data from that station, and therefore a robust comparison is only possible for the first sampling campaign. For this reason, we decided to focus further analyses exclusively on the GSRS data while retaining the comparison with Ny-Ålesund as a meaningful reference point in the discussion of spatial variability.
- R3 I suggest to follow a more schematic approach to discuss the variability of elements/ions in snow across seasons and years. I would start distinguishing (statistically of course) those species which present always the same concentration from those presenting significant differences. I would apply this approach considering the whole years and single seasons (a comparison between the 3 winters considers, 3 springs and so on). In this way it would be possible to clearly highlight where you should focus to explore a correlation between snow composition and meteo/climatic conditions. At this point it would be rather simple to apply some statistical tool to assess whether snow data are correlated to meteorological variables.
- A3. The authors appreciate Reviewer 2's valuable suggestion. Considering that the Reviewer 2 has evaluated a previous version of the manuscript, this aspect was already improved in the second version thanks to the previous reviewers. However, to better address this point, we first applied the robust, non-parametric Kruskal-Wallis test to assess whether significant differences existed among seasons. This was followed by the post-hoc Dunn test to identify which specific seasons differed, with significance evaluated using adjusted p-values (< 0.05). The results, presented at the end of # 3.2, are reported in Table S5 and Fig. S3. Finally, we performed a Spearman correlation on the selected variables (the ones that showed a significant variability across seasons and years) to examine potential relationships between snow composition and meteorological/climatic conditions. The correlations between the analysed species (ions and trace elements) and the meteo/climatic variables were generally weak, with all coefficients below 0.4. The weak correlations indicate a complex influence of the considered meteorological variables on the analysed species concentrations, suggesting that the effects of Arctic amplification may involve additional or interacting processes that require additional deep investigations.

- **R4** In general the meteorological description of the three years considered in the study is rather qualitative and not very well supported by data. You report many passages like "Lower surface air temperatures, reduced precipitations, higher wind speed (m sec-1), and minor mean snow height with respect to the typical AA conditions, were induced by strong cold polar vortex triggered by a net positive Arctic Oscillation (AO) phase, and recorded in the 2019-20 winter season". It would be more valuable if you could **provide some number (mean temperatures, anomalies, deviations) to provide a quantitative framework** to support your description.
- **A4.** We thank the Reviewer for raising this aspect. We first opted for putting meteorological information in Table S3 (cited in the text), but we saw the Reviewer's point and we provided quantitative information to support the description in the main text (L324-329).
- R5 About Pb: you make some discussion about the period presenting anomalously high Pb concentration, assuming an enhanced atmospheric transport from pollution sources (related to the weakening of the polar vortex) as the most likely cause. It would be interesting to see the behavior of other element typically related to atmospheric pollution to see if something similar is observe (As, Zn, and others, you measured a lot!).
- A5. We appreciate the Reviewer's suggestion to explore the behaviour of other elements typically associated with atmospheric pollution (such as As, Zn, and others). The relationship between Pb and other anthropogenic species is recognized with HCA where As and Zn have been shown short linkage distance. We have chosen to focus primarily on Pb in this manuscript due to its clear role in the observed anomalous concentration period. Moreover, adding other species in Figure 3, already very complex, would generate more confusion without adding new significant information. Given the already discussed tracers of atmospheric transport, we feel that the current discussion provides already significant insights into the potential link between enhanced pollution sources and the meteorological-climatic conditions.
- **R6** You refer to many Figures reported in the Supplementary material, while in the end you only included a few figures in the main text. This is not very helpful as the reader has continuously to shift from the main text to the supplement. I suggest to **move some of the supplementary figures in the main text** and eventually to limit the number of figures in the supplementary. **A6.** We see the Reviewer's point, and we moved Figure S1 to the main text (now Figure 1), after modifying it as suggested by Reviewer's 1. We also decided to add a new Fig. 5 (correlation plots) in the manuscript, in spite of placing it in the SI.

Considering the above, I can't recommend this manuscript for publication at this stage. From my point of view major revisions are needed.

Please find below some more specific comments:

R7 - Figure 1: I can't see solar radiation, but this is reported in the caption. Please add something to indicate the years shown in the graphs.

A7. Actually, in the last version of the manuscript, we didn't report solar radiation in the caption. Caption claims: "Figure 1. AO Index, air temperature (°C), precipitation (mm), snow height (cm),  $\Delta h$  snow height (cm), wind speed (m sec-1), and wind direction (°) from the NCEP/NCAR Reanalysis data. The green horizontal line above the wind speed graph indicates the 5 m sec-1 threshold, above which wind drift may occur on surface snow layers. The colour of the line refers to the  $\Delta h$  color scale, which indicates negative values of  $\Delta h$ , NOAA Physical Sciences Lab's daily composites tool was used to calculate the near-surface air temperatures across the Northern Hemisphere from October to May. Grey bands indicate the winter periods."

We also reported the years in x-axis already.

**R8 - Figure 3** and related discussion about clustering: I can't see in the list all the elements that you measured (as reported in the introduction of the manuscript. For example Zn).

**A8.** Also in this case, we already modified the Figure during the first round of revisions. The referee 2 evaluated the previous version of the manuscript.

**R9 - Line 103:** "this is the snow layer most influenced by the aerosol-cryosphere exchanges" not very clear, maybe "this is the snow layer most affected by exchanges involving snow, atmosphere and aerosol"?

**A9.** We see the point of Reviewer2, and we agreed to slightly change this sentence. We rephrased as follows: "this is the snow layer most impacted by aerosol deposition and exchange processes at the snow-atmosphere interface".

**R10 - Line 208-215:** it is not clear the reason why you choose to apply the test. You want to highlight the difference among two different data populations right? Maybe it would be better to clearly report what the two populations are.

**A10.** We appreciate the reviewer's comment. To clarify, our aim was indeed to compare the ionic loads at Ny-Ålesund and Gruvebadet, which represent the two data populations under study. We applied the Wilcoxon rank-sum test (Mann-Whitney U test) to assess whether there was a statistically

significant difference in contamination levels between these two sites. We revised the manuscript accordingly, to state this more clearly.

- **R11 Line 231:** it would be interesting to know what ions showed significant differences among the two trends and provide a little bit of discussion about this.
- **A11.** Wilcoxon test highlighted significant differences among the two populations of samples (NyÅ and GSRS) only for Na+, K+, NO3-, Br-, and MSA, which are all marine species, likely most influenced by the closeness of NyÅ site to the coast.
- R12 Line 238-239: what % data refer to? To total ionic load?
- **A12.** % refer to the concentrations, not the ionic loads.
- R13 Line 277: it is clear what you want to say, but "Arctic type" is not very rigorous. I suggest to rephrase
- **A13.** We appreciate the Reviewers' feedback. However, we already clarified this aspect during the previous round of revisions, and no more "Arctic type" expressions have been used within the main text.
- **R14 Line 295-296:** but you said before that sulphates are not always dominated by ss fraction, so I would add something to say that sulphate are not always dominated by marine aerosol.
- **A14.** Also in this case, we already clarified this aspect during the previous round of revisions.
- R15 Line 304-306: how the presence of drift ice can explain a higher deposition of geogenic elements? The thing is clear for marine-related species but I can't understand for the geogenic ones.
- A15. We thank the Reviewer for this insightful comment. While statistical analysis did not show significant differences for the geogenic elements (Al, Ca, Mn, Sr), except for Fe, which weakly correlated with air temperature, we hypothesize that their increase may still be linked to the overall atmospheric conditions during the late spring 2020 period, although correlations may not have been captured fully in the statistical analysis. For example, these conditions could have promoted the resuspension of dust and sediment from the local environment, such as from coastal areas, which could have been transported by stronger winds to the study area. To avoid overinterpretation of the text, we decided to modify this paragraph as follows "These conditions likely enhanced the production of sea spray aerosols, which, when carried by winds, may have increased the deposition of marine species onto the snowpack. The increased deposition of geogenic elements might also have

been influenced by low temperature anomalies (as seen with Fe), and/or by stronger wind speeds, although significant correlations have not emerged in this preliminary statistical analysis."

---

## Author Response (AR3)

**Reviewer #4**

In this updated version of the manuscript the authors have taken into account most of the comments raised by reviewers, with evident improvements.

Despite the general quality of the manuscript has greatly improved, I still have a few comments that the authors can find below.

R1 - To compare the two series measured at the two considered sites (Ny-Ålesund and GSRS site) the authors apply the Wilcoxon test. From what I know, the use of the Wilcoxon signed-rank test does not appear appropriate in your case, since the two datasets were collected over different time periods and with different temporal resolutions. The signed-rank test requires paired observations (i.e., measurements taken simultaneously or under directly comparable conditions). You also report this at line 245 (matched or dependent observations). In your context, the samples should be treated as independent, and a test designed for unpaired data (e.g., the Mann–Whitney U test) would be more suitable. But maybe this observation is only due to a lack of details in your manuscript. I suggest to apply a different test or to provide further information.

**A1** – We see the raised point, and we thank the Reviewer for this suggestion. After a thorough evaluation, this comparison was excluded, and the Authors decided to focus only on GSRS data, as kindly suggested. The whole manuscript was then revised accordingly.

**R2** - Another thing that I don't really understand about the application of this test is found in line 277, where you report that the test showed a good agreement between the two series except for some sporadic peaks. The test is applied to whole data-series, not to single data points. How is it possible that the test is working good in general but with the exception of these sporadic peaks? Please provide some more detail about this.

**A2** – We agree with the Reviewer. The test was applied to whole data-series, not to single data points. We missed to well-explain this part. What the Authors intended to say here was that the Wilcoxon test showed a general good agreement for most of the investigated species. However, from a visual comparison, sporadic peaks showed minor discrepancies between the two series. In any case, as previous declared, the Authors agreed to focus only on GSRS data, removing the GSRS-NyÅ series comparison.

R3 - In my first review I was wondering whether focusing only on GSRS would have been better to build a more concise and direct manuscript. The authors replied that the comparison was important as it serves to evaluate the local variability of the considered variables. According to this I still don't completely understand the reason for this comparison. The authors made a comparison (even if maybe

not with the best statistical test); they found that for some variables there is no difference, for some variables there is some difference; in light of this they decide to focus only on GSRS site. This is not very logical to me. The test highlights that for some species there is a significant difference, but the authors do not discuss this. They just say that according to the results they decide to focus only on GSRS. So I don't understand how the evaluation of local variability was carried out. The test shows that for some species there is a significant local variability, but this is never discussed across the manuscript. Maybe I am missing something, but according to this, I recommend again not to include Ny-Alesund data in the manuscript and focus only on GSRS.

**A3** – The Authors thanks the Reviewer for raising this point and agreed to not include the NyÅ-GSRS comparison.

**R4** - A lot of information frequently cited in the main text is still present only in the supplementary. I suggest to include in Table 1 not only seasonal mean values, but also yearly mean values. This would require to add only a few additional rows to the table and make reading easier. Please provide p-values for the determined correlation coefficients (lines 350-352).

**A4** – According to the Reviewer's suggestion, the Authors have added yearly mean values to Table 1 for completeness. However, due to the dynamic conditions at the GSRS site, such as accumulation, melting, and wind drift, these yearly averages may be particularly subjected to spikes. The "Total" reported in Table 1 slightly changes compared to the former Table 1, as the Authors noticed a copyand-paste mistake from the original table in excel. Previous "Total" was indeed referred to the sum of the most (>1%) and less (<1%) abundant species in the surface snow.

Additionally, the Authors have updated the following statement "The results indicated generally weak positive correlations ( $\rho < 0.5$ ) for the variables considered." at lines 303-304 by adding "( $\rho < 0.5$ , p-values < 0.05)", as kindly suggested. P-values were not repeated for all the variables as they were all < 0.05, and this was stated before citing variables one-by-one.

**R5** - In the end the correlation study between glaciochemistry and meteorological variables did not bring interesting results. From what I see the composition of snow does not seem to be correlated significantly with meteorological data. According to this I would smooth a bit the conclusions, removing all the discussion related to the Arctic Amplification (lines 640-643). In the end your interpretation shows a link between sea ice dynamics and snow composition, but it does not highlight any correlation between climatology and snow chemistry (also because the investigated time interval is too short).

**A5** – The Authors agreed with this suggestion and smoothed the conclusions accordingly. The discussion related to the Arctic Amplification was thus removed.

According to the above, I think that once the authors will adjust their manuscript considering these comments, the manuscript will be ready for publication.

---

## Author Response (AR4)

**Editor decision: Publish subject to minor revisions (review by editor)**

**ANSWERS TO EDITOR'S COMMENTS**

I carefully read your paper and also in view of your responses to previous referee reports that were not rereviewed by the referees. I came to the conclusion that it still needs further revision. Due to the numerous changes, your paper contains now several inconsistencies and also lacks essential information that was included in your responses to referees but is not included in the paper. Please thoroughly read the paper and **check for logical structure**, **completeness in terms of essential information and quality standards in terms of scientific and presentation quality.** Please consider my comments below as exemplary; if you see additional opportunities to improve the paper, I will highly appreciate it. After careful revision, I will be happy to accept it for publication.

We thank the Editor for her time and suggestions, and we answer all the comments one-by-one, revising the manuscript accordingly.

**I. Context and implications**

- E1) Upon the comments by referee #2 you had already toned down the possible role of Arctic Amplification. Further Referee #4 also asked you to remove wording about AA But yet, your abstract prominently starts with 'Arctic Amplification' implying that this is the topic of the paper. In the remainder of the manuscript, Arctic Amplification is only mentioned at the very end of Sect 4 in a rather vague and general sentence where I even wonder whether AA could be replaced by 'interannual variability'. If AA is the main topic of the paper, it should be discussed and also show up in the discussion and conclusion sections; if not, the abstract should be modified such that it properly introduces the topic.
- A1) We thank the Editor for this valuable suggestion. Our initial intention in mentioning Arctic Amplification (AA) was to provide a broader scientific framework: while AA is not the focus of our study, it constitutes the background in which Arctic interannual variability takes place. However, we acknowledge that this framing may have given the impression that AA was a central topic of the manuscript, which is not the case. To avoid any ambiguity and to better reflect the actual scope of our work, we have now removed all references to AA from the abstract and the main text, and revised the manuscript accordingly.
- E2) l. 383 389: In which figures can these trends be observed? Please elaborate this discussion (also considering my initial comment regarding AA above). If these are key findings from your analysis, they deserve a clearer description and the 'valuable insights' should be more clearly discussed.
- A2) We appreciate the Editor's suggestions, and we decided to add the appropriate references to the correspondent figures and tables for major clarity. The mention to AA was removed, and the

discussion about "valuable insights" has been extended as follows: "These contrasting patterns suggest that variability in atmospheric circulation and oceanic state exerts a direct control on the timing and intensity of aerosol deposition to the snowpack. In particular, stronger winds and warmer air masses enhance the transport and deposition of crustal and sea-salt species, whereas negative oceanic anomalies appear to modulate their availability as sources. This highlights how combined shifts in atmospheric and oceanic conditions drive the observed interannual variability of ionic and elemental concentrations in surface snow."

**E3) Conclusions:**

The conclusions can still be improved: Structure them according to our author guidelines (see below) and pay particular attention to the last point. Currently, the main conclusions are mostly related to snow composition. Of course, snow composition is related to atmospheric composition, but it is not explicitly stated here.

ACP guidelines for the title, abstract, and concluding section: <a href="https://www.atmospheric-chemistry-and-physics.net/policies/guidelines">https://www.atmospheric-chemistry-and-physics.net/policies/guidelines</a> for authors.html

A3) We thank the Editor for this kind suggestion, and we agreed to improve the conclusions following the guidelines for concluding section. Conclusions were modified including main quantitative results (e.g., "the highest concentrations of marine species recorded in late spring 2020 (e.g., Cl- = 110 mg m-2, Na+ = 52 mg m-2, SO42- = 28 mg m-2). These elevated levels coincided with the most extensive sea ice in Kongsfjorden in March 2020 (FI = 113.28 km2, DI = 16.53 km2),"), and providing a comparison with previous studies to put them in context (e.g., "[...]as also indicated by elevated Brenr mean values (17.7 compared to the usual < 1; Barbaro et al., 2021)", "[...] This approach mitigates potential biases for EFs results that may come from reference element selection, assumptions about crustal composition, and background variability (Reimann and de Caritat, 2000)."). Caveats and limitations were clearly stated (e.g., "Limitations such as weak correlations between meteorological variables and concentrations suggest that further multivariate and long-term analyses are needed to quantify these relationships more robustly."), and implications were concisely presented.

**II. Referee comments & responses**

- E4) Several pieces of information that is in the reply to reviewers has not been implemented into the text. Examples are included below, but I ask you to carefully go through all responses and see what essential information may be interesting for the reader:
- A4) We thank the Editor for raising up this point. We carefully checked all the replies to reviewers and we think the most crucial part are now reported within the manuscript.

In the main text, we added this part, which was clarified answering to R-46 (Referee#1) in l. 677-680: "[...] a Hierarchical Cluster Analysis (HCA) method was carried out using the whole dataset, which encompasses all three sampling campaigns (2018-2021) to capture the full variability in sulphate sources and atmospheric processes across different seasons and years."

Additionally, we added this clarification (answer to R-17, Referee#1) in l. 256-258: "Average loads are presented in place of medians to avoid underrepresentation of the occasional high concentration events, which are critical for understanding the snowpack chemistry dynamics in the Arctic environment."

and this sentence in l. 553-555 "The coefficient 0.12 is based on the molar ratio of SO42- to Na+, while the value 0.175 represents the observed slope corresponding to the pre-industrial nss-SO42-/nss-Ca ratio in mineral dust found in snow (Schwikowski et al., 1999, and reference therein)."

E4.1: Sampling strategy of collecting the upper 3 cm as pointed out by Referee #2. There is a lengthy discussion in your response – however, this seems important information and its essence should be moved to the text or at least into a separate supplemental section, referenced in the main text.

A4.1) We changed this part as follows, trying to capture the main essence of the discussion with Referee #2:

l. 97-108 "[...] Each sample was collected 10 cm apart from the previous one, along a precise path. This method was designed to minimise the temporal variability between consecutive samples and reduce the impact of potential spatial variability (within the 5-15% range, according to Spolaor et al., 2019).

During the first sampling campaign, carried out from October 4th, 2018 to May 10th, 2019, 133 snow samples were collected at the Gruvebadet Snow Research Site (GSRS), a clean-area located about 1 km south of Ny-Ålesund, nearby the Gruvebadet Atmospheric Laboratory (GAL), dedicated to the chemical and physical monitoring of the seasonal snowpack (Scoto et al., 2023; Fig. 1).

The surface snow was sampled within the upper 3 cm, as this layer is the most directly affected by atmospheric deposition and snow-atmosphere exchanges (Spolaor et al., 2018, 2021b). Sampling only the uppermost layer reduces the risk of signal dilution in deeper snow layers, and ensures a simple, robust protocol suitable for long-term campaigns with minimal disturbance of the snowpack."

E4.2: "A more focused and contextualized discussion is necessary to justify the broader implications claimed by the authors."

Yet, the abstract only states "By comparing the snow chemical composition of the 2019-20 season with 2018-19 and 2020-21, we provide insights into the interplay between short-term meteorological variability and the long-term climatic impacts of AA in Svalbard, as well as associated shifts in aerosol production process."

A4.2: We removed any AA mentions in the abstract and within the whole text. These sentences were thus removed "By comparing the snow chemical composition of the 2019-20 season with 2018-19 and 2020-21, we provide insights into the interplay between short-term meteorological variability

and the long-term climatic impacts of AA in Svalbard, as well as associated shifts in aerosol production process." and modified as follows "This study provides a detailed characterisation of how snow chemistry in this area responds to sea ice extent, atmospheric circulation, and broader Arctic climatic variability."

**III. Structure**

E5) Sections 4.1 – 4.3 could still use some restructuring or even different titles. It seems that section 4.1 "Ny-Ålesund seasonal and interannual trends variability in surface snow "could end in line 392. Then a new section could start about 'enrichment factors' that could first properly introduce this factor (how is it defined?) with two subsections 'ER in surface snow' (l. 398 – 409) and 'Main ion sources and ER in seasonal snow' (current sect 4.2). Section 4.3, solely dedicated to ER of Br, seems not correctly weighted, i.e. it should be a subsection to one the previous sections. If you disagree, please make the current structuring clearer by either changing section titles or clarifying why the text as is belongs into the sections as indicated.

- A5) We carefully went through these sections and we retained they could be changed as follows, to maintain the proper logical flow:
- 4.1 Ny-Ålesund seasonal and interannual trends variability in surface snow removing the Pb discussion
- 4.2 Enrichment Factors and source attribution with a brief description of what EFs are
- 4.2.1 Enrichment Factors in seasonal snow of Ny-Ålesund presenting EFs analyses results
- 4.2.2 Main ion sources in seasonal snow of Ny-Ålesund leaving this part as it was
- 4.2.3 Bromine enrichment

**IV. Technical Comments**

- E6) 1.90: Define MSA here, at its first use
- A6) We thank the Editor for raising this point. We defined MSA as its first use, as suggested.

**E7) l. 345: What are 'wintry concentrations'? It sounds rather colloquial.**

A7) We see the point of the Editor and we agreed to change this expression to "wintertime concentrations".

E8) Fig 2: Please improve the figure caption. Ideally, add a, b, c, d to the panels and describe them accordingly. Right now it is rather confusing for the reader, e.g. "The green horizontal line above the wind speed graph" implies that the green line is located in the panel above the wind graph plot. "The colour of the line refers 375 to the  $\Delta h$  color scale, which indicates negative values of  $\Delta h$ " – do you mean the color of the green line? If so, can't you just state the value?

A8) We added the panels to the figure, and we discussed them in the figure caption, as suggested by the Editor. We changed the figure caption as follows: "Fig. 2. a) AO Index; b) air temperature (°C); c) precipitation (mm); d) snow height (cm) and  $\Delta h$  snow height (cm); e) wind speed (m sec-1) and wind direction (°) from the NCEP/NCAR Reanalysis data. The green horizontal line in panel e) indicates the 5 m sec-1 threshold, above which wind drift may occur on surface snow layers. The colour of this line refers to the  $\Delta h$  color scale shown in panel d), where green indicates negative values of  $\Delta h$ . NOAA Physical Sciences Lab's daily composites tool was used to calculate the near-surface air temperatures across the Northern Hemisphere from October to May. Grey bands across the panels indicate the winter periods."

E9) Figure 3: Improve the caption. At a minimum, describe the panels from top to bottom (right now, you first explain the bottom plot, followed by the top left, moving eventually to the right ...). Explain all species, incl Br(enr). Why is the windspeed plot shown twice?

*A9)* We accept the Editor's request and we improved the caption of Fig. 3, modifying it as follows "Ionic loads (mg m-2) of Na+, Cl-, Mg, SO42-, nss-SO42-, MSA, Br-, Brenr, Ca, Sr, Mn, Fe, Al, Pb, V, Ni in the surface snow for the three sampling campaigns: 2018-19, 2019-20, 2020-21. Seasonal trends are here presented for specific elements to provide a detailed view of how concentrations vary across distinct sampling periods. Precipitation trends (mm), wind speed (m sec-1), and wind direction (°) are reported twice for visual clarity."

The panels were arranged in two columns, separating marine species from crustal and anthropogenic ones. Instead of describing the panels strictly from top to bottom, we chose to group together species showing similar trends or peaks, as this allowed us to provide a more cohesive and integrated discussion of the results. We therefore believe that a purely sequential description of the panels would not effectively convey the relationships among the different species.

Wind speed, together with wind direction and precipitations, is shown twice for visual clarity, considering the Referee #1 suggestions:

 R-5 I suggest the concentration time series be plotted along with the precipitation data and wind direction. That way the reader can directly match fresh precipitation and sea salt (or crustal) input.) - R-30 I.373-375: trace elements (and ions) concentration time series would be best plotted with wind direction and speed.

**E10) Fig 5: Improve the figure caption such that its content is self-explanatory. Are these empty cells for MSA in 2019-2020? If so why?**

A10) We changed the figure caption as follows, for clarity: "Fig. 5. Spearman correlation plots. a) Correlation plot for the three sampling seasons; b) Correlation plot for the 2018-2019 season; c) Correlation plot for the 2019-2020 season; d) Correlation plot for the 2020-2021 season. Higher positive correlations are presented in shades of red, while lower negative correlations are shown in shades of blue."

For the 2019-2020 correlation plot, the "empty" cells referred to MSA were values approaching to zero, as indicated by the previous legend. However, for clarity, we have now changed the colour scale.

**E11) l. 521 - 526: try to break these very long sentences in shorter ones or remove unnecessary information.**

A11) We thank the Editor for raising this point. We are not sure the Editor was referring to 1.521-526 or the following paragraph. However, we changed both as follows, breaking these very long sentences in shorter ones: "Contrarily, in the 2018-19 season, sea ice lasted only until April and was restricted to the inner, shallower parts of Kongsfjorden (Assmy et al., 2023). This limited duration may not have provided enough time with adequate sunlight for substantial biological activity to accumulate beneath or within the ice. This occurred despite the dominance in 2019, unlike the following year, of *Phaeocystis pouchetii*, a phytoplankton species known for its capacity to generate DMS in significant quantities (Assmy et al., 2023).

Finally, the ammonium (NH4+) load showed positive correlations with several ions (Fig. 5). It was strongly correlated with Na+ ( $\rho_{load} = 0.71$ , p-value < 0.05), Cl- ( $\rho_{load} = 0.52$ , p-value < 0.05) and K+ ( $\rho_{load} = 0.72$ , p-value < 0.05). Positive correlations were also observed with SO42- ( $\rho_{load} = 0.54$ , p-value < 0.05), NO3- ( $\rho_{load} = 0.45$ , p-value < 0.05), MSA ( $\rho_{load} = 0.37$ , p-value < 0.05) and Br- ( $\rho_{load} = 0.53$ , p-value < 0.05). These results suggest a close link with sea-salt ions and biogenic emissions. However, some contribution from biomass burning events and potential influence from anthropogenic activities cannot be excluded."